# NirD curtails the stringent response by inhibiting RelA activity in *Escherichia coli*

**Loïc Léger[1], Deborah Byrne[2], Paul Guiraud[1], Elsa Germain[1], Etienne Maisonneuve[1]***

[1]Laboratoire de Chimie Bactérienne, Institut de Microbiologie de la Méditerranée, CNRS-Aix Marseille Univ (UMR7283), Marseille, France; [2]Protein Expression Facility, Institut de Microbiologie de la Méditerranée, CNRS-Aix Marseille Univ, Marseille, France

**Abstract** Bacteria regulate their metabolism to adapt and survive adverse conditions, in particular to stressful downshifts in nutrient availability. These shifts trigger the so-called stringent response, coordinated by the signaling molecules guanosine tetra and pentaphosphate collectively referred to as (p)ppGpp. In *Escherichia coli*, accumulation of theses alarmones depends on the (p)ppGpp synthetase RelA and the bifunctional (p)ppGpp synthetase/hydrolase SpoT. A tight regulation of these intracellular activities is therefore crucial to rapidly adjust the (p)ppGpp levels in response to environmental stresses but also to avoid toxic consequences of (p)ppGpp over-accumulation. In this study, we show that the small protein NirD restrains RelA-dependent accumulation of (p)ppGpp and can inhibit the stringent response in *E. coli*. Mechanistically, our in vivo and in vitro studies reveal that NirD directly binds the catalytic domains of RelA to balance (p)ppGpp accumulation. Finally, we show that NirD can control RelA activity by directly inhibiting the rate of (p)ppGpp synthesis.

## Introduction

Bacteria have evolved numerous molecular mechanisms to detect and cope with environmental stress, including the use of nucleotide-based signaling pathways to efficiently coordinate cellular processes and provide a fast response. Among these signaling pathways, the stringent response is a general stress response that is mediated by the accumulation of the nucleotides guanosine 5′-diphosphate 3′-diphosphate (ppGpp) and guanosine 5′-triphosphate 3′-diphosphate (pppGpp), collectively known as (p)ppGpp (*Potrykus and Cashel, 2008*). These alarmones allow bacteria to rapidly respond and adapt to various conditions of nutritional and environmental stress by affecting gene expression and metabolism. In Gram-negative bacteria, (p)ppGpp binds RNA polymerase, thereby altering its promoter selectivity, which results in genome-wide transcriptional reprograming (*Durfee et al., 2008*; *Ross et al., 2013*; *Ross et al., 2016*; *Sanchez-Vazquez et al., 2019*; *Traxler et al., 2008*). Additionally, (p)ppGpp also binds directly and alters the activity of several enzymes including DNA primase, translation factors, lysine decarboxylase, and polyphosphate kinase (*Corrigan et al., 2016*; *Kanjee et al., 2012*; *Wang et al., 2019a*; *Zhang et al., 2018*). Importantly, this rewiring of cell physiology also appears to play a critical role in the regulation of bacterial virulence, survival during host invasion, and antibiotic resistance and tolerance (*Hauryliuk et al., 2015*; *Hengge, 2020*; *Irving et al., 2021*; *Potrykus and Cashel, 2008*).

In *Escherichia coli*, (p)ppGpp synthesis and hydrolysis are regulated by RelA and SpoT, which are members of RelA/SpoT homologue (RSH) (*Atkinson et al., 2011*) enzymes and share similar domain architecture. The enzymatic N-terminal half (NTD) has the synthetase (SYN) and the hydrolase (HYD) domains, and the C-terminal half (CTD) of the protein contains conserved domains, which play a critical role in sensing and transducing stress signals to the catalytic domains (*Gratani et al., 2018*;

*For correspondence:
emaisonneuve@imm.cnrs.fr

**Competing interests:** The authors declare that no competing interests exist.

*Hauryliuk et al., 2015*; *Hogg et al., 2004*; *Mechold et al., 2002*; *Pausch et al., 2020*; *Takada et al., 2020*; *Tamman et al., 2020*). SpoT is a bifunctional enzyme with both hydrolytic and synthetic activities while RelA is a monofunctional (p)ppGpp synthetase. Indeed, despite close sequence similarity between RelA and SpoT, RelA maintains a 'pseudo'-hydrolase domain that is structurally conserved but enzymatically inactive. The regulation of RelA and SpoT activities represents an obvious checkpoint for (p)ppGpp levels which reveals many complexities (reviewed in *Irving and Corrigan, 2018*; *Ronneau and Hallez, 2019*).

Activation of the (p)ppGpp synthetase RelA occurs via a ribosomal mechanism in response to amino acid starvation or other stressful conditions, which ultimately result in amino acid starvation, for example, fatty acid starvation, that leads to lysine starvation (*Haseltine and Block, 1973*; *Sinha et al., 2019*). Under this condition, deacylated tRNAs accumulate and RelA interacts with uncharged tRNA at the vacant ribosomal A-site causing the activation of its (p)ppGpp synthetic activity (*Arenz et al., 2016*; *Brown et al., 2016*; *Loveland et al., 2016*; *Winther et al., 2018*). Inversely, RelA is suggested to be inhibited by intra- and/or inter-molecular interactions under replete conditions (*Gropp et al., 2001*; *Turnbull et al., 2019*; *Yang and Ishiguro, 2001*), even though it retains residual activity (*Murray and Bremer, 1996*). Importantly, the hydrolysis function of SpoT provides an opposing activity that is crucial for balancing the basal activity of RelA. Indeed, disruption of the *spoT* gene results in a toxic accumulation of (p)ppGpp and is therefore lethal (*Xiao et al., 1991*).

The regulation of SpoT activities involves sources of nutritional stress other than amino acid starvation. These include the deficiency of fatty acid (*Seyfzadeh et al., 1993*), phosphate (*Spira et al., 1995*), carbon (*Xiao et al., 1991*), and iron (*Vinella et al., 2005*). Interaction of SpoT with other factors seems to regulate the catalytic balance between the synthetic and hydrolytic activities. During fatty acid starvation, the accumulation of (p)ppGpp required a specific interaction between the acyl carrier protein and SpoT (*Battesti and Bouveret, 2006*). Similarly, the binding of the protein YtfK with the catalytic domains of SpoT triggers (p)ppGpp accumulation during phosphate or fatty acid starvation (*Germain et al., 2019*). In addition, stimulation of the SpoT hydrolase activity is driven through direct interaction with the anti-$\sigma^{70}$ factor Rsd upon carbon downshift (*Lee et al., 2018*).

Given the importance of (p)ppGpp in stress survival, virulence, and antibiotic tolerance, we have designed a genetic assay for the identification of new protein candidates that can modulate (p)ppGpp homeostasis in *E. coli*. We thereby discovered that overproduction of NirD, the small subunit of the nitrite reductase, decreases intracellular levels of (p)ppGpp. Importantly, our results show that overexpression of *nirD* limits RelA-dependent accumulation of (p)ppGpp in vivo and can prevent activation of the stringent response during amino acid starvation. Moreover, our results show that NirD is not involved in (p)ppGpp breakdown. Rather we convincingly show that, mechanistically, NirD inhibits the (p)ppGpp synthetase activity of RelA through a specific and direct interaction with the catalytic domains of RelA in vivo and in vitro.

## Results

### Identification of genes counteracting the toxicity of (p)ppGpp over-accumulation mediated by RelA

At high concentrations, (p)ppGpp becomes a potent inhibitor of bacterial cell growth. Indeed, rising (p)ppGpp levels by overexpression of the *relA* gene has been shown to inhibit *E. coli* growth in amino acid-rich media (*Schreiber et al., 1991*). To gain further insights into (p)ppGpp homeostasis, we developed a screening assay to identify genes that in multiple copies could reverse this toxicity (*Figure 1A*). In this study, the toxic accumulation of (p)ppGpp is achieved by induction of the *relA* gene cloned in the low-copy-number plasmid pBbS2k under the control of the anhydrotetracycline (aTc)-inducible $P_{tet}$ promoter (*Figure 1A*). *E. coli* MG1655 cells harboring the pBbS2k-*relA* were transformed with a pooled collection of plasmids obtained from the ASKA library containing almost all *E. coli* K-12 genes (*Kitagawa et al., 2005*), each cloned into the high-copy-number vector pCA24N downstream of the isopropyl β-D-1-thiogalactopyranoside (IPTG)-inducible $P_{T5-lac}$ promoter (*Figure 1A*). This genetic screen led us to identify 88 clones that repetitively grow on non-permissive conditions. Sequence analysis of the pCA24N derivatives from these 88 clones enabled the identification of four different genes. The *spoT*, *gppA*, and *nirD* genes were predominant with each one

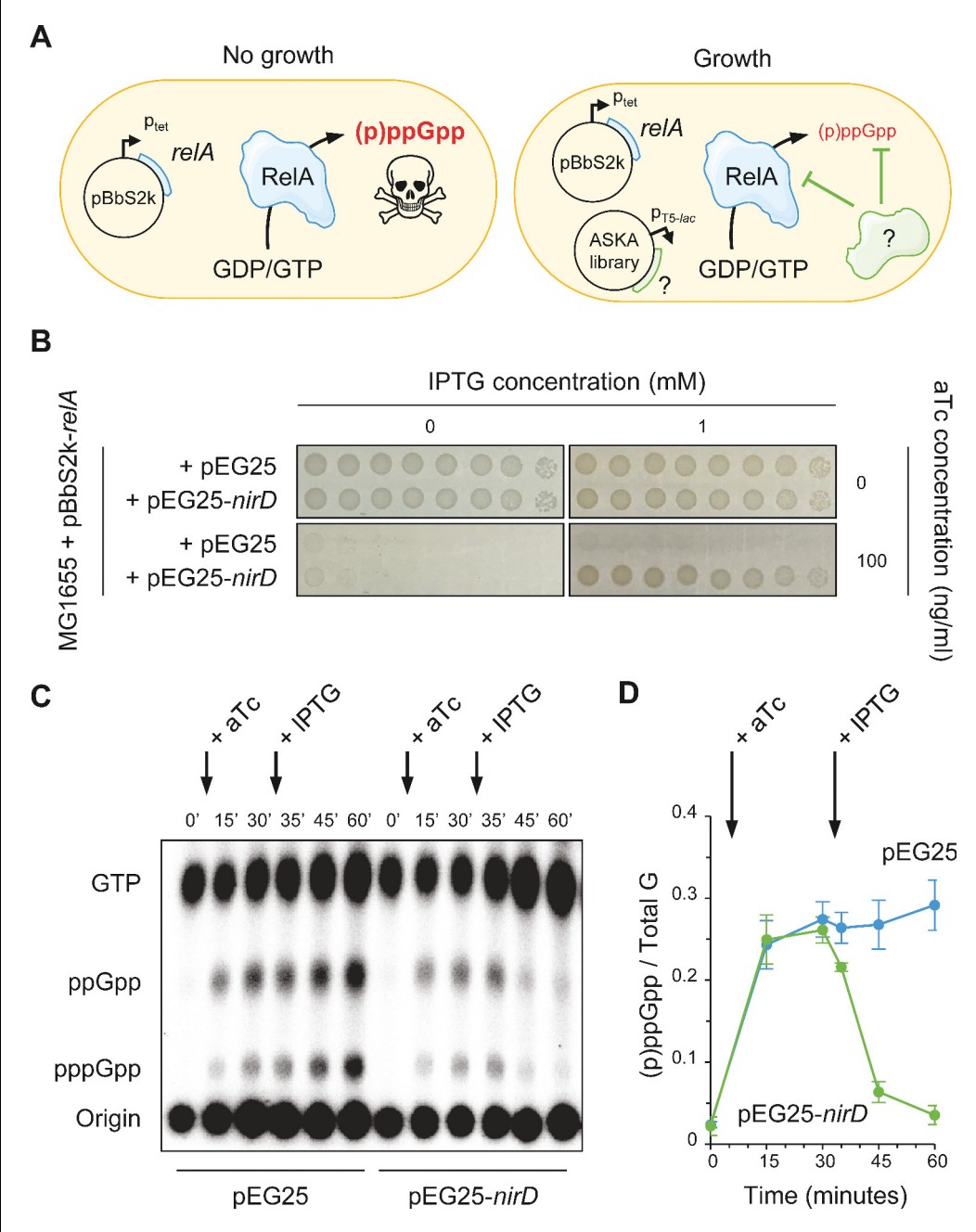

**Figure 1.** NirD can limit the RelA-dependent accumulation of (p)ppGpp. (**A**) Schematic representation of the genetic assay using ASKA library to identify genes that in multiple copies would suppress the growth defect associated with toxic over-accumulation of (p)ppGpp. Question marks denote a putative candidate that limits RelA-dependent accumulation of (p)ppGpp. (**B**) NirD suppresses the growth defect associated with RelA overproduction. Wild-type *E. coli* MG1655 cells were co-transformed with the plasmid derivatives pBbS2k-*relA* and pEG25-*nirD* inducible by anhydrotetracycline (aTc) and isopropyl β-D-1-thiogalactopyranoside (IPTG), respectively. Serial dilutions of stationary-phase cultures were spotted on NA plates containing the indicated concentrations of inducers. Additional controls and inducer concentrations are shown in *Figure 1—figure supplement 1A*. The results are representative of three independent experiments with similar results. (**C, D**) In vivo (p)ppGpp dynamic in *E. coli* MG1655 cells carrying pBbS2k-*relA* and pEG25-*nirD* after consecutive expression of the *relA* and *nirD* genes using 100 ng/mL aTc and 1 mM IPTG, respectively. Nucleotides extracted from samples collected at the indicated times were separated by thin layer chromatography. The autoradiogram (**C**) is representative of three

*Figure 1 continued on next page*

*Figure 1 continued*

independent experiments, and the curves of the relative levels of (p)ppGpp (D) are represented as the means of the three independent experiments, the error bars depicting the SDs.

The online version of this article includes the following source data and figure supplement(s) for figure 1:

**Source data 1.** Raw autoradiogram.
**Source data 2.** Quantification of (p)ppGpp.
**Figure supplement 1.** NirD can inhibit RelA-dependent (p)ppGpp synthesis.

accounting for almost a third of the genes carried by the sequenced plasmids (33.0, 36.4, and 29.5%, respectively) while *uhpA* was only identified once (1.1%). Our observations that *spoT* and *gppA* revert toxicity associated to (p)ppGpp accumulation is consistent with previous observations (*Germain et al., 2019*; *Murray and Bremer, 1996*; *Sanyal and Harinarayanan, 2020*). Considering the *gppA* and *spoT* genes as internal controls of the screen, we further focused our work on *nirD*.

## NirD balances (p)ppGpp level produced by RelA

*nirD* encodes the small subunit of the cytoplasmic nitrite reductase (*Harborne et al., 1992*; *Peakman et al., 1990a*) and has never been linked to the (p)ppGpp metabolism. We first confirmed the result obtained with the high-copy-number vector pCA24N-*nirD* from the ASKA library by re-cloning the coding region of *nirD* (untagged) in a more suitable physiological plasmid harboring a tight IPTG-inducible $P_{T5-lac}$ promoter (pEG25). Indeed, as shown in *Figure 1B*, NirD can robustly overcome the toxicity associated to (p)ppGpp over-accumulation mediated by RelA. To further assess the role of NirD in the stringent response, we monitored (p)ppGpp levels in vivo after consecutive inductions of RelA and NirD in a double expression system assay. As shown in *Figure 1C, D*, induction of RelA is associated with a massive accumulation of (p)ppGpp, which is sharply stopped and then decreased upon NirD induction.

The rapid (p)ppGpp decay observed following NirD induction raises the possibility that NirD may trigger SpoT-dependent (p)ppGpp hydrolysis. However, NirD still suppresses toxicity associated to RelA overproduction when the double expression assay experiment is performed in a Δ*relA spoT* background (*Figure 1—figure supplement 1A*). These results show that the hydrolase activity of SpoT is not involved in the reduced (p)ppGpp accumulation observed upon NirD induction. Importantly, overexpression of *nirD* in a Δ*relA* strain failed to phenocopy the well-known amino acid auxotrophic characteristic of a ppGpp$^0$ strain (Δ*relA spoT* mutant) (*Xiao et al., 1991*), strongly arguing that NirD does not act as a small alarmone hydrolase (*Figure 1—figure supplement 1B*). Taken together, our results show that NirD impairs RelA-dependent (p)ppGpp synthesis in *E. coli*.

## NirD can prevent the stringent response in *E. coli*

Since NirD can efficiently overcome toxicity associated to artificial level of (p)ppGpp when RelA is overproduced (*Figure 1B*), we next address whether similar effect is observed during physiological condition known to activate RelA. In natural environment, RelA synthesizes the alarmones in response to amino acid deprivation. Indeed during amino acid starvation, deacylated tRNAs accumulate and RelA interacts with uncharged tRNA at the vacant ribosomal A-site causing the activation of its (p)ppGpp synthetic activity (*Arenz et al., 2016*; *Brown et al., 2016*; *Loveland et al., 2016*; *Winther et al., 2018*). RelA activity is therefore required to survive under amino acid starvation, and as a result a Δ*relA* mutant is not able to grow in the presence of 1 mM serine, methionine, and glycine (SMG), which induces isoleucine starvation (*Uzan and Danchin, 1978*). Interestingly, we observed that overexpression of *nirD* in wild-type (WT) strain phenocopies the growth defect of a Δ*relA* mutant on SMG plate (*Figure 2A*), suggesting that *nirD* also prevents (p)ppGpp accumulation from endogenous RelA under these conditions. To further address this possibility, we monitored the dynamics of (p)ppGpp levels under amino acid starvation upon the addition of L-valine (which also induces isoleucine starvation). As shown in *Figure 2B, C*, and as previously observed (p)ppGpp rapidly increases in WT strain and reaches maximum levels within 5 min (*Cashel and Gallant, 1969*). Remarkably, when NirD is induced prior to amino acid starvation, no significant accumulation of (p)ppGpp is detected (*Figure 2B, C*). Taken together, our results show that once induced NirD abolishes the stringent response in *E. coli*.

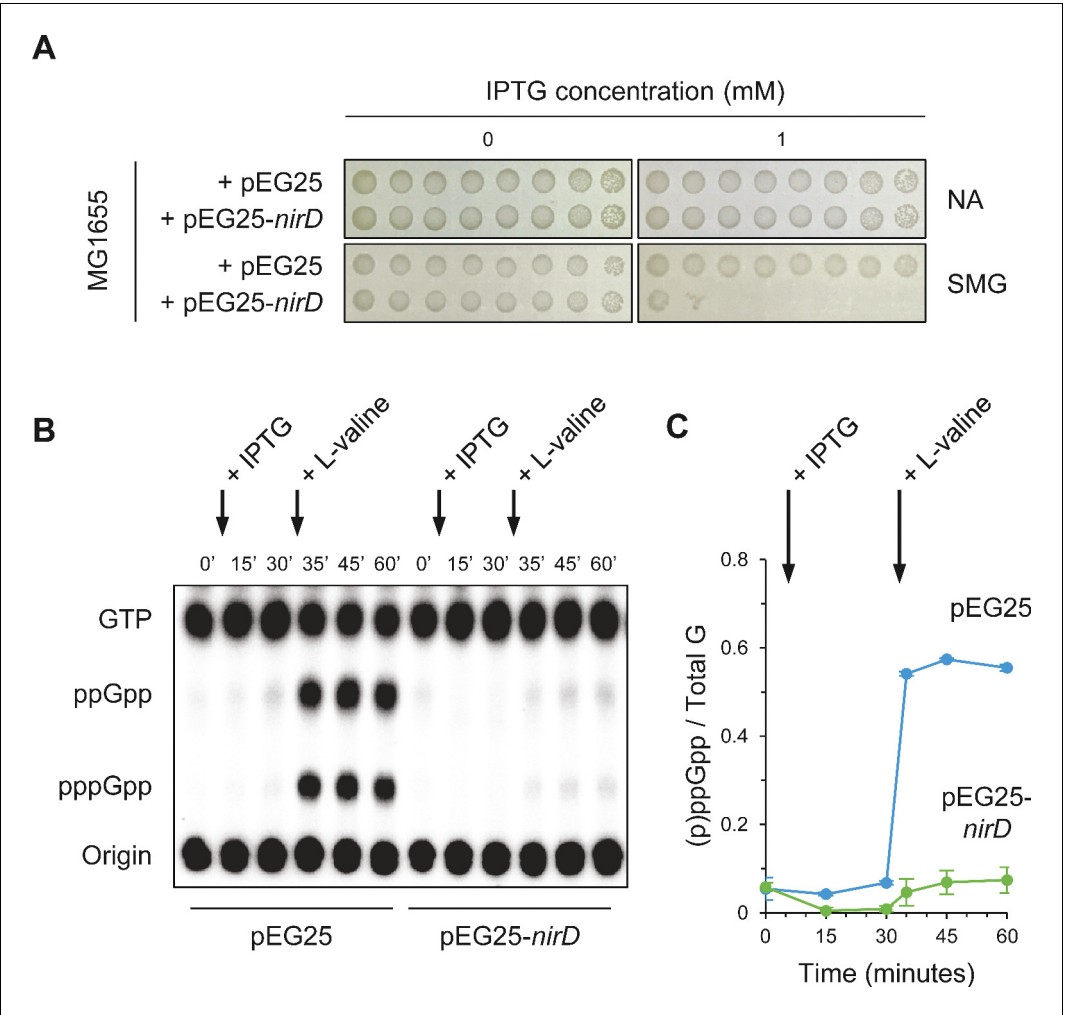

**Figure 2.** NirD inhibits accumulation of (p)ppGpp upon amino acid starvation. (**A**) NirD inhibits cell growth on M9-glucose minimal medium supplemented with serine, methionine, and glycine (SMG). Stationary-phase cultures of wild-type *E. coli* MG1655 strain harboring pEG25-*nirD* or pEG25-*spoT* were spotted on NA and SMG plates containing the indicated concentrations of isopropyl β-D-1-thiogalactopyranoside (IPTG). The results are representative of three independent experiments. (**B, C**) Exponentially growing cells of wild-type *E. coli* MG1655 harboring pEG25 or pEG25-*nirD* were challenged for amino acid starvation by addition of 500 μg/mL of L-valine. *nirD* is induced by the addition of 1 mM of IPTG. Nucleotides were extracted and separated by thin layer chromatography. The autoradiogram (**B**) is representative of three independent experiments, and the curves of the relative levels of (p)ppGpp (**C**) are represented as the means of the three independent experiments, the error bars depicting the SDs.

The online version of this article includes the following source data for figure 2:

**Source data 1.** Raw autoradiogram.
**Source data 2.** Quantification of (p)ppGpp.

## NirD functionally interacts with RelA in vivo

Regulation of RSH activities through protein–protein interactions has been previously reported (*Battesti and Bouveret, 2006*; *Germain et al., 2019*; *Hahn et al., 2015*; *Karstens et al., 2014*; *Krüger et al., 2020*; *Lee et al., 2018*; *Peterson et al., 2020*; *Raskin et al., 2007*; *Ronneau et al., 2016*; *Ronneau et al., 2019*; *Wout et al., 2004*). We therefore tested whether NirD is able to interact with RelA in vivo using a bacterial two-hybrid (BTH) assay (*Karimova et al., 1998*). For that purpose, the complementary T18 and T25 domains of *Bordetella pertussis* adenylate cyclase (CyA) were fused to the N-terminus of NirD and RelA proteins using the two compatible vectors pUT18C and

pKT25, respectively. When the *E. coli* BTH101 strain (*cya* deficient) is co-transformed with plasmids harboring T18 and T25 fusions, NirD exhibited a strong interaction in vivo with RelA (*Figure 3A*). However, despite the strong sequence homologies between the paralogous RelA and SpoT proteins (*Metzger et al., 1989*), no physical interaction is observed between SpoT and NirD in vivo by BTH assay (*Figure 3A*). This result therefore supports a strong specificity of the interaction between NirD and RelA.

To further assess the significance of the NirD-RelA interaction in vivo and its role in (p)ppGpp homeostasis, we randomly mutagenized *nirD* and searched for defective mutants unable to interact with RelA. One of such mutants that had charge reversal amino acid substitution, glutamate 50 to lysine (E50K), displayed compromised interaction with RelA (*Figure 3A*). Importantly, we confirm that both T18-NirD and T18-NirD$^{E50K}$ recombinant proteins were correctly expressed and folded as shown by their ability to strongly interact with the catalytic subunit of the nitrite reductase NirB (*Figure 3—figure supplement 1*) (*Harborne et al., 1992*). Importantly and on the contrary to the results obtained with the WT copy of *nirD*, we observed that overexpression of *nirD*$^{E50K}$ is not able to suppress toxicity associated to RelA overproduction on plate (*Figure 3B*). This result predicts that breaking the interaction with RelA abolishes the inhibitory effect of NirD on (p)ppGpp accumulation. Indeed, as shown in *Figure 3C, D*, *nirD*$^{E50K}$ failed to stop the massive accumulation of (p)ppGpp observed in vivo after RelA overproduction as compared to the fast decay of (p)ppGpp observed when the WT copy of *nirD* is expressed (*Figure 3C and D*).

*nirD* is a second gene of a four-gene operon (*nirBDC-cysG*) strongly induced under anaerobic growth (*Harborne et al., 1992*; *Peakman et al., 1990b*). Interestingly, we observed that Δ*nirD* mutant presents an extended growth recovery in anaerobic SMG liquid medium consistent with a toxic over-accumulation of (p)ppGpp in this strain (*Figure 3E*). Indeed, a chromosomal encoded *nirD*$^{E50K}$ mutant, which is functional for the nitrite reductase activity (*Figure 3—figure supplement 1C*) but failed to interact with RelA, displays the same growth defect, suggesting that the NirD-RelA interaction is important to adjust proper (p)ppGpp level under this specific growth condition. Taken together, our results show that a specific functional interaction between RelA and NirD can lower RelA-dependent accumulation of (p)ppGpp in vivo.

## NirD targets the catalytic N-terminal region of RelA

RelA consists of several protein domains that have been encompassed in two functional regions, the catalytic NTD and the regulatory CTD. The N-terminal region includes the enzymatically inactive (p) ppGpp hydrolase domain (HYD) as well as the functional (p)ppGpp synthetase domain (SYN), while the CTD comprises the Thr-tRNA synthetase, GTPase and SpoT domain (TGS), the alpha helical domain (AH), the zinc finger domain (ZFD), and finally the RNA recognition motif domain (RRM) (*Figure 4A*) (*Brown et al., 2016*; *Loveland et al., 2016*). To determine which of these domains are required for the interaction with NirD, we assayed the interaction between NirD and several truncated RelA proteins using the BTH assay.

Truncated RelA proteins were fused to the T25 domain of the *B. pertussis* adenylate cyclase and transformed together with NirD fused to the T18 domain in a *cya*-deficient *E. coli* strain (BTH101). BTH analysis revealed that NirD interacts with the RelA$^{N-terminal}$ fusion comprising the catalytic domains (*Figure 4B*). Moreover, careful quantification of β-galactosidase activity revealed that none of the four domains encompassing the CTD of RelA is required for this interaction in vivo. However, truncated RelA fusion lacking either the synthetase domain (RelA$^{1-181}$) or the 'pseudo' hydrolase domain (RelA$^{181-744}$) failed to interact in vivo with NirD. Therefore, the catalytic domains of RelA seem necessary and sufficient to interact with NirD in vivo. The specificity of the interaction between NirD and RelA was further reinforced by the absence of interaction observed between NirD and SpoT$^{N-terminal}$ as well as NirD$^{E50K}$ and RelA$^{N-terminal}$ (*Figure 4—figure supplement 1A*). Interestingly, the C-terminal region of RelA is proposed to play a key role in the autoregulation of the enzymatic activity, and as a result the RelA$^{N-terminal}$ truncated protein is constitutively active (*Schreiber et al., 1991*). Given that NirD interacts with RelA$^{N-terminal}$, we therefore tested the intriguing possibility that NirD may also be able to inhibit this dysregulated variant. Remarkably, as shown in *Figure 4—figure supplement 1B*, we observed that NirD but not NirD$^{E50K}$ inhibits the toxicity associated with (p) ppGpp over-accumulation mediated by the constitutively active RelA$^{N-terminal}$. Taken together, our results show that NirD can inhibit RelA activity by interacting with its catalytic N-terminal region in vivo.

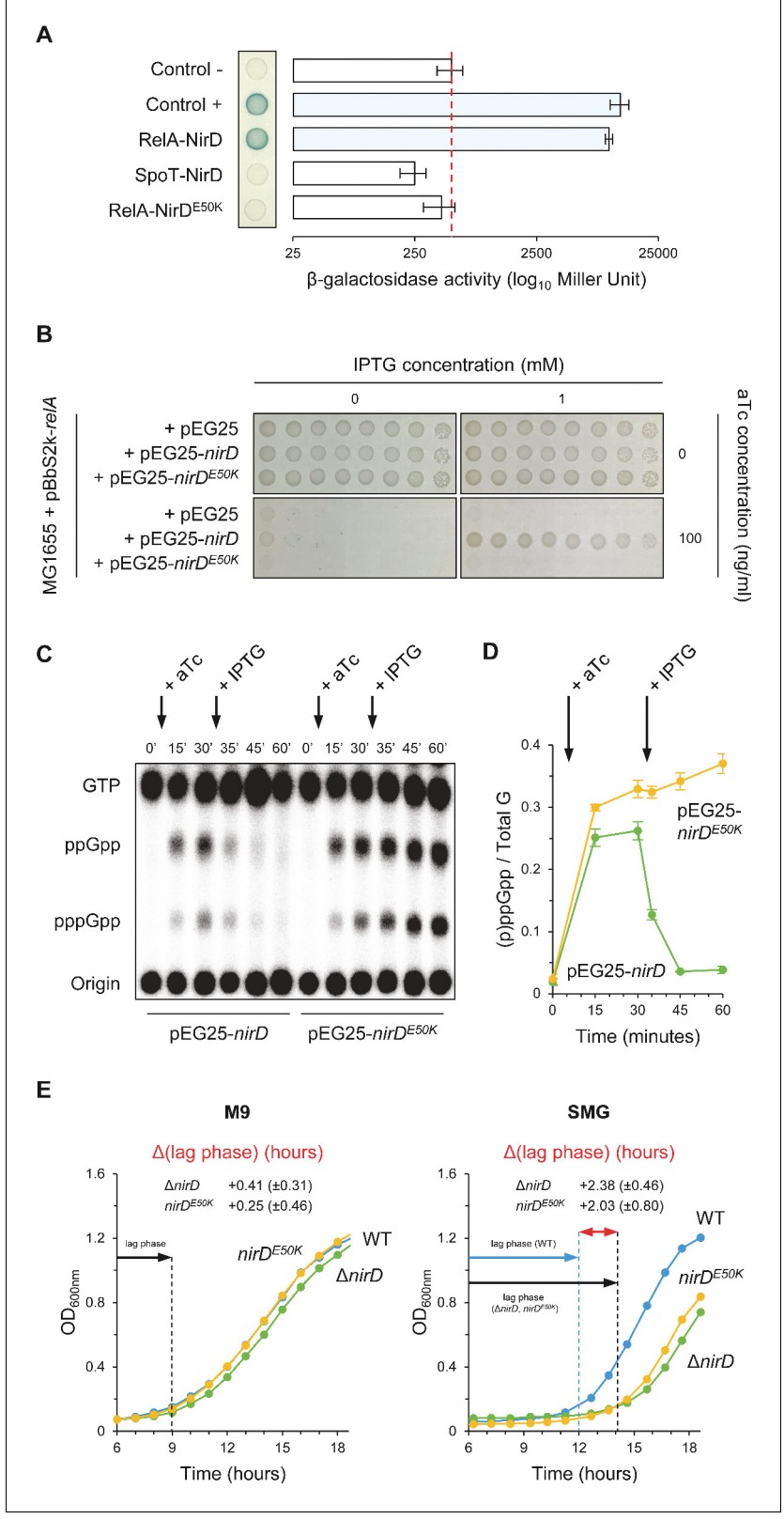

**Figure 3.** NirD can inhibit RelA-dependent accumulation through physical interaction in vivo. (**A**) Bacterial two-hybrid assay using *E. coli* BTH101 cells co-transformed with plasmid derivatives pKT25-*relA* or pKT25-*spoT* and pUT18C-*nirD* or pUT18C-*nirD*^E50K^. Stationary-phase cultures were spotted on NA plates containing X-Gal as a blue color reporter for positive interaction. The bars showing β-galactosidase activity are represented as the means of
*Figure 3 continued on next page*

*Figure 3 continued*

three independent experiments, and the error bars depict the SDs. (B) Growth assay of *E. coli* MG1655 cells co-transformed with the plasmid derivatives pBbS2k-*relA* and pEG25-*nirD* or pEG25-*nirD*^E50K^ inducible by anhydrotetracycline (aTc) and isopropyl β-D-1-thiogalactopyranoside (IPTG), respectively. Serial dilutions of stationary-phase cultures were spotted on NA plates containing the indicated concentrations of inducers. The results are representative of three independent experiments. (C, D) In vivo (p)ppGpp level dynamic in *E. coli* MG1655 cells carrying pBbS2k-*relA* and pEG25-*nirD* or pEG25-*nirD*^E50K^ after successive inductions of expression of the *relA* and *nirD* or *nirD*^E50K^ genes using 100 ng/mL aTc and 1 mM IPTG, respectively. Nucleotides extracted from samples collected at the indicated times were separated by thin layer chromatography. The autoradiogram (C) is representative of three independent experiments, and the curves of the relative levels of (p)ppGpp (D) are represented as the means of the three independent experiments, the error bars depicting the SDs. (E) Growth curves of wild-type (WT) cells, Δ*nirD*, and *nirD*^E50K^ mutants under anaerobiosis in M9-glucose minimal medium without amino acids (left panel) or supplemented with 1 mM serine, methionine, and glycine (SMG) (right panel) (see Materials and methods). The growth curves are representative of at least three independent experiments, and the Δ(lag phase) are represented as the means (± SD) of the independent experiments.

The online version of this article includes the following source data and figure supplement(s) for figure 3:

**Source data 1.** Determination of β-galactosidase activity.
**Source data 2.** Raw autoradiogram.
**Source data 3.** Quantification of (p)ppGpp.
**Source data 4.** Determination of Δ(lag phase).
**Figure supplement 1.** NirD^E50K^ functionally interacts with NirB.
**Figure supplement 1—source data 1.** Determination of β-galactosidase activity.
**Figure supplement 1—source data 2.** Raw data for growth curves.

## NirD directly binds RelA to inhibit the rate of (p)ppGpp synthesis in vitro

To confirm the in vivo interaction of NirD at the catalytic N-terminal region of RelA, we used different complementary in vitro methods. For that purpose, the RelA catalytic domains (RelA^N-terminal^), NirD, and NirD^E50K^ were individually produced and purified by two consecutive chromatographies, affinity and size-exclusion (*Figure 5—figure supplement 1A–C*). We first performed size-exclusion chromatography (SEC) experiment separating RelA^N-terminal^ and NirD alone or pre-mixed. As shown in *Figure 5—figure supplement 1D*, co-elution resulted in an earlier eluting peak, suggesting the formation of a stable complex. Indeed, SDS-PAGE analysis confirmed that the two proteins were co-eluted from the column (*Figure 5—figure supplement 1E*).

To further characterize the NirD-RelA^N-terminal^ interaction, we then used biolayer interferometry (BLI), an in vitro approach for measuring biomolecular interactions. Biotinylated NirD was immobilized on streptavidin biosensors as ligand, and RelA^N-terminal^ was used as the analyte. By adding RelA at concentrations ranging from 0.1 to 30 μM, a dose-response association is recorded and decreased during the dissociation step corresponding to the washing of the sensor (*Figure 5A*), showing that NirD directly interact with the catalytic N-terminal region of RelA. Moreover, by plotting the maximum binding response against the corresponding concentrations of RelA, we calculated a dissociation constant ($K_D$) of 0.55 μM (± 0.053 μM) (*Figure 5A*). In addition, and in agreement with the BTH assays (*Figure 4—figure supplement 1A*), RelA^N-terminal^ did not interact with the NirD^E50K^ variant in vitro (*Figure 5A*). Finally, to further determine the thermodynamic parameters of the NirD-RelA^N-terminal^ interaction, we performed isothermal titration calorimetry (ITC) experiment. The assay first confirmed the direct interaction between the two proteins with an estimated $K_D$ of 0.34 μM (*Figure 5B*), very similar to that estimated by BLI (*Figure 5A*). This reaction is a favorable spontaneous exothermic reaction with a ΔH kcal/mol of −3.63 ± 0.208 and is enthalpy driven at 25°C. On the contrary and consistent with BLI experiments, no heat exchange was recorded upon titration of RelA^N-terminal^ by NirD^E50K^, further supporting the strong specificity of the interaction between NirD and RelA (*Figure 5B*). Collectively, these results show that NirD interacts directly and specifically with the catalytic domains of RelA in vitro.

Finally, these results are in line with our in vivo results showing that NirD inhibits accumulation of (p)ppGpp from the constitutively active RelA^N-terminal^ protein (*Figure 4—figure supplement 1B*) and therefore support a rather simple model where NirD would directly inhibit RelA synthetic activity

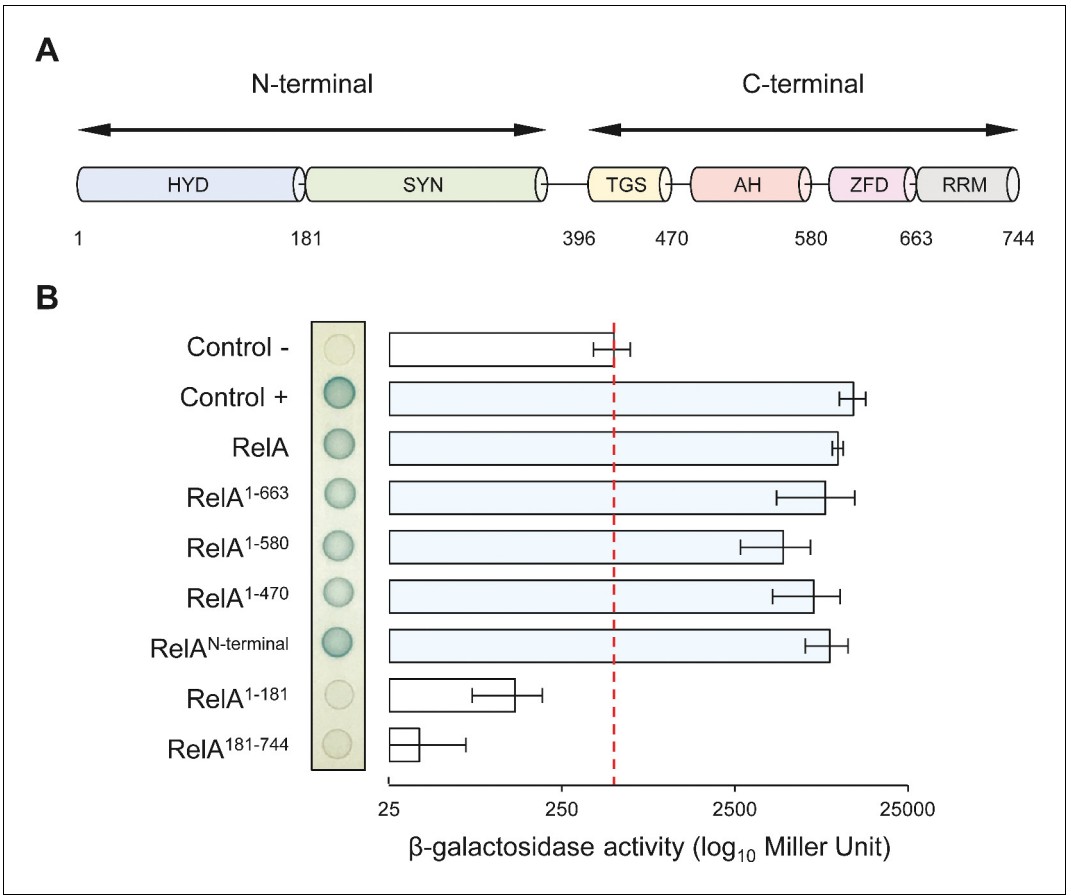

**Figure 4.** NirD can interact with the catalytic N-terminal region of RelA. (**A**) Schematic representation of RelA and its protein domains. (**B**) Bacterial two-hybrid assay using *E. coli* BTH101 cells co-transformed with plasmid derivatives pUT18C-*nirD* and pKT25 with the full-length or truncated *relA* gene as indicated. Stationary-phase cultures were spotted on NA plates containing X-Gal as a blue color reporter for positive interaction. The bars showing β-galactosidase activity are represented as the means of three independent experiments, and the error bars depict the SDs.

The online version of this article includes the following source data and figure supplement(s) for figure 4:

**Source data 1.** Determination of β-galactosidase activity.

**Figure supplement 1.** The RelA$^{N-terminal}$-NirD interaction appears specific.

**Figure supplement 1—source data 1.** Determination of β-galactosidase activity.

through direct binding. To challenge this hypothesis, we performed in vitro pppGpp synthesis assay with purified proteins. We observed that RelA$^{N-terminal}$ protein could efficiently synthetize [$^{32}$P]-pppGpp, but the addition of NirD strongly affected the synthetase activity in vitro. Indeed, 80% of GTP are converted to pppGpp within 60 min by RelA, whereas only 30% are converted when NirD is added (*Figure 6A, B*). Importantly, addition of the NirD$^{E50K}$ does not affect the RelA synthetic activity (*Figure 6A, B*). Finally, careful examination of the initial rates (calculated over the first 150 s) shows a concentration-dependent inhibition of RelA's (p)ppGpp synthetic activity by NirD with a half-inhibition occurring at ~1:1 molar ratio and a near complete inhibition for the highest tested concentration of NirD (1:16 molar ratio) (*Figure 6C*). Taken together, these results show that NirD can block pppGpp synthetase activity of RelA through a direct interaction with the catalytic domains.

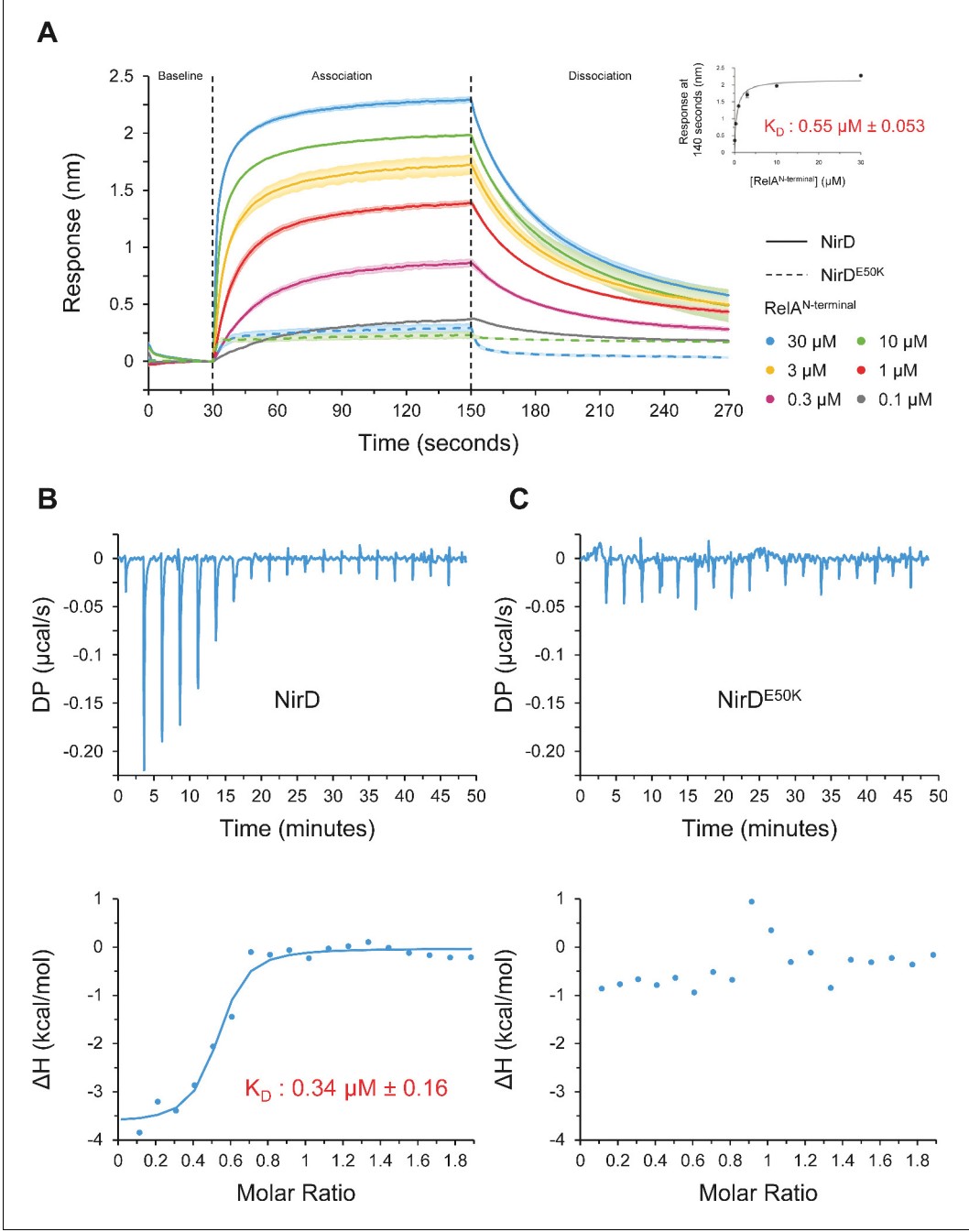

**Figure 5.** NirD directly interact with the catalytic N-terminal region of RelA. (**A**) Biolayer interferometric assay of RelA$^{N\text{-terminal}}$ on NirD or NirD$^{E50K}$. Biotinylated NirD or NirD$^{E50K}$ were immobilized on streptavidin biosensors and probed with RelA$^{N\text{-terminal}}$ at concentrations ranging from 0.1 to 30 μM. The curves are represented as the means of the subtracted reference binding responses during association and dissociation from three experiments, the error bars depicting the SDs. The inset curve shows the specific biolayer interferometry (BLI) response (nm) 10 s before the end of association as a function of RelA$^{N\text{-terminal}}$ concentration. Data are represented as the means of the three experiments, the error bars depicting the SDs. (**B, C**) Isothermal titration calorimetry profiles corresponding to the binding of RelA$^{N\text{-terminal}}$ 30–300 μM NirD (**B**) or 300 μM NirD$^{E50K}$ (**C**) at 25°C. The upper panels show raw data for titration of NirD or NirD$^{E50K}$ with RelA$^{N\text{-terminal}}$, and the lower panels show the integrated heats of binding obtained from the raw data. The data were fitted using a 'One Set of Sites' model in the PEAQ-ITC Analysis Software.

The online version of this article includes the following source data and figure supplement(s) for figure 5:

*Figure 5 continued on next page*

*Figure 5 continued*

**Source data 1.** Raw data for biolayer interferometry (BLI) experiments.

**Source data 2.** Raw data for isothermal titration calorimetry experiment run with NirD.

**Source data 3.** Raw data for isothermal titration calorimetry experiment run with NirD$^{E50K}$.

**Figure supplement 1.** NirD can directly interact with the catalytic N-terminal region of RelA.

**Figure supplement 1—source data 1.** Raw data for purification of RelA$^{N-terminal}$.

**Figure supplement 1—source data 2.** Raw SDS-PAGE for RelA$^{N-terminal}$ purification.

**Figure supplement 1—source data 3.** Raw data for purification of NirD.

**Figure supplement 1—source data 4.** Raw SDS-PAGE for NirD purification.

**Figure supplement 1—source data 5.** Raw data for purification of NirD$^{E50K}$.

**Figure supplement 1—source data 6.** Raw SDS-PAGE for NirD$^{E50K}$ purification.

**Figure supplement 1—source data 7.** Raw data for size-exclusion chromatography experiment separating RelA$^{N-terminal}$ and NirD alone or pre-mixed.

**Figure supplement 1—source data 8.** Raw SDS-PAGE for size-exclusion chromatography experiment.

## Discussion

*E. coli* contains two long RSH enzymes, RelA and SpoT, that synthesize the alarmones (p)ppGpp. While SpoT is a bifunctional enzyme, RelA is monofunctional with a degenerated hydrolase domain, making SpoT the primary source of hydrolysis (*Xiao et al., 1991*). Therefore, a fine-tune regulation of the intracellular (p)ppGpp synthetic and hydrolytic activities in bacteria is crucial to rapidly adjust (p)ppGpp level in response to environmental stress but also to prevent toxic consequences due to (p)ppGpp over-accumulation (*Xiao et al., 1991*). Over the past decade, several molecular mechanisms regulating the activity of (p)ppGpp synthetase and/or hydrolase have been characterized (*Battesti and Bouveret, 2006*; *Germain et al., 2019*; *Hahn et al., 2015*; *Karstens et al., 2014*; *Krüger et al., 2020*; *Lee et al., 2018*; *Peterson et al., 2020*; *Raskin et al., 2007*; *Ronneau et al., 2016*; *Ronneau et al., 2019*; *Wout et al., 2004*). In this study, we uncovered an unprecedented mode of regulation that can prevent accumulation of (p)ppGpp in *E. coli*. This regulation implicates physical interaction in vivo and in vitro between the catalytic domains of RelA and the protein NirD, the small subunit of the cytoplasmic NADH-dependent nitrite reductase complex NirBD (*Harborne et al., 1992*). We show that NirD can functionally and directly inhibit the RelA (p)ppGpp synthetic activity. RelA and SpoT are paralogous proteins; however, despite strong sequence homologies and similar domain architecture we were not able to notice physical interaction between NirD and SpoT (*Figure 3A* and *Figure 4—figure supplement 1A*), supporting a specific physiological role of NirD in inhibiting RelA synthetic activity. The absence of a functional link between NirD and SpoT is further reinforced by the observation that NirD does not seem to lower basal (p)ppGpp level produced by the weak (p)ppGpp synthetase activity encoded by SpoT in the absence of RelA (*Figure 1—figure supplement 1B*).

Stimulation of RelA synthetase activity in *E. coli* is driven by stalled ribosomes upon amino acid starvation (*Haseltine and Block, 1973*; *Wendrich et al., 2002*; *Winther et al., 2018*). In addition, NtrC-dependent transcription of *relA* takes an important role in (p)ppGpp accumulation (*Brown et al., 2014*). Therefore, the identification of a potent intracellular inhibitor of RelA (p)ppGpp synthetase represents an additional level of complexity in the regulation to the current stringent model mediated by RelA. Our results show that the NirD protein level can play an important role for adjusting intracellular (p)ppGpp levels. Indeed, ectopic induction of NirD is sufficient to prevent accumulation of (p)ppGpp mediated by RelA in the absence of nutritional stress (*Figure 1C, D*). Moreover, we show that once expressed, *nirD* can also totally abolish activation of the stringent response during amino acid starvation (*Figure 2C, D*). The estimated concentration of RelA is in the range of 100–400 nM in *E. coli* (*Li et al., 2014*; *Pedersen and Kjeldgaard, 1977*). Importantly, we found in vitro that the NirD-RelA affinity is characterized by a $K_D$ in this concentration range (300–550 nM) (*Figure 5A, B*). Therefore, at concentrations above the $K_D$, NirD would interact with RelA to catalytically inhibit its activity. *nirD* is a second gene of a four-gene operon (*nirBDC-cysG*) expressed under anaerobic growth (*Harborne et al., 1992*; *Peakman et al., 1990b*) and maximally induced in the presence of glucose and nitrate (*Tyson et al., 1997*; *Wang and Gunsalus, 2000*). Under these conditions, NirD is predicted to reach μM range (*Khlebodarova et al., 2016*) and regulation of RelA

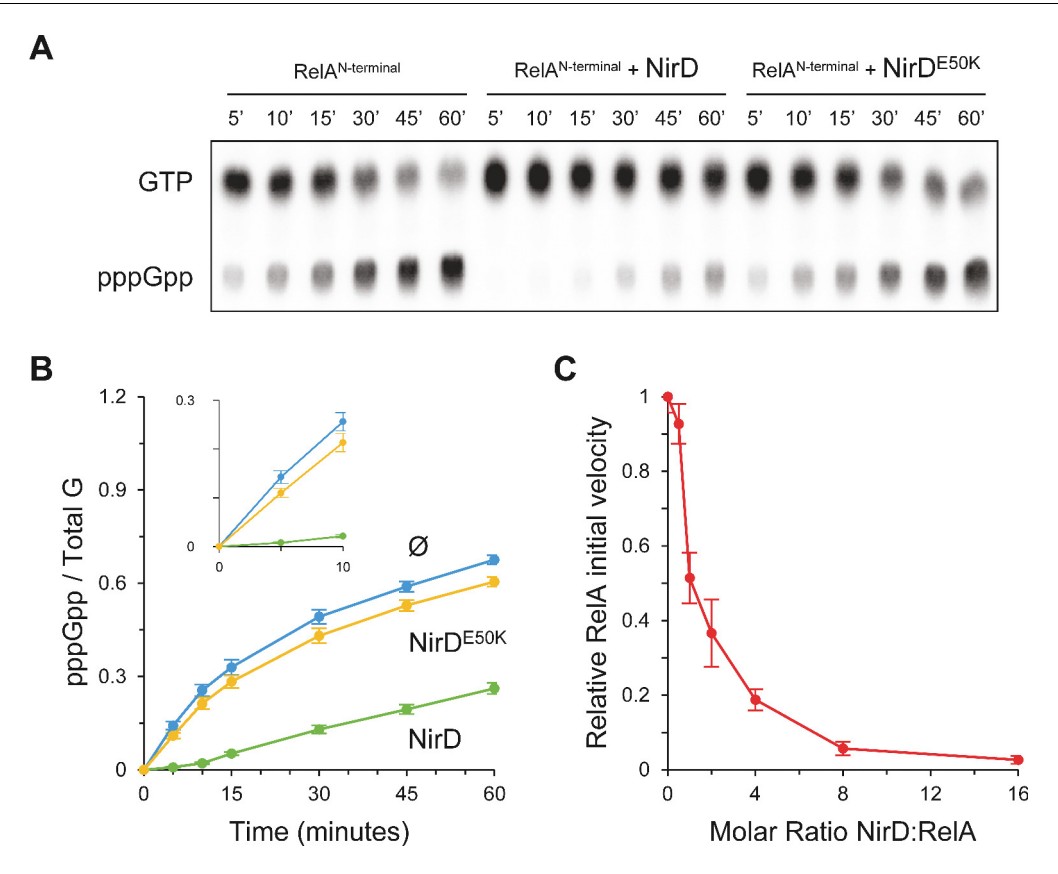

**Figure 6.** NirD inhibits RelA activity in vitro. (**A, B**) In vitro pppGpp formation by 1 µM RelA[N-terminal] in the presence or absence of 4 µM NirD or NirD[E50K]. Samples collected at the indicated times were separated by thin layer chromatography (TLC). The autoradiogram (**A**) is representative of three experiments, and the curves of the relative levels of pppGpp (**B**) are represented as the means of the three experiments, the error bars depicting the SDs. (**C**) Relative initial velocity of RelA as a function of NirD:RelA molar ratio. The in vitro formation of pppGpp by 1 µM RelA[N-terminal] was assayed in the presence or absence of NirD at concentrations ranging from 0.5 to 16 µM. Samples were collected every 30 s over a period of 150 s and separated by TLC to determine the relative initial velocity (estimated by linear regression using five data points). Results are represented as the means of three experiments, and the error bars depict the SDs.

The online version of this article includes the following source data for figure 6:

**Source data 1.** Raw autoradiogram.
**Source data 2.** Quantification of pppGpp.
**Source data 3.** Determination of the relative initial velocity.

by NirD would therefore become physiologically relevant. Importantly, while NirB and NirD can form a stable complex needed for the NADH-dependent nitrite reductase activity, we observed that coexpression of *nirB* and *nirD* can still counteract RelA-dependent (p)ppGpp accumulation (*Figure 3—figure supplement 1B*). The cytoplasmic nitrite reductase NirBD has emerged as a key regulator of the nitric oxide (NO) level during nitrate respiration (*Bulot et al., 2019*). Indeed, by reducing nitrite generated during nitrate reduction, NirBD would limit the level of toxic nitrite and thus its further reduction to NO (*Bulot et al., 2019*). In addition, our results present evidence that NirD can also promote bacterial fitness by adjusting (p)ppGpp levels in response to amino acid starvation under anaerobic glucose fermentation, a condition where the nitrite reductase is expressed but where the substrate is missing (*Figure 3E*). This regulation is dependent on the RelA-NirD interaction since NirD[E50K], which is not affected for the nitrite reductase activity (*Figure 3—figure supplement 1C*) but failed to interact with RelA (*Figure 3A*), does not enhance bacterial fitness in response to amino acid

starvation under anaerobic glucose fermentation (*Figure 3E*). Therefore, our results suggest that NirD can link (p)ppGpp homeostasis to anaerobic metabolism in *E. coli*. Additional analyses are currently underway to further investigate the physiological roles of the RelA-NirD interaction in *E. coli*.

The CTD of RSH enzymes is assumed to play key roles in sensing nutrient starvation and adjusting the enzymatic state of the NTD. In that sense, the RelA structures on stalled ribosomes emphasized the role of CTD in sensing nutrient availability. Once bound to the ribosome, RelA adopts an open extended conformation, which is thought to release the autoinhibitory effect of the CTD on the NTD, therefore promoting (p)ppGpp synthesis (*Arenz et al., 2016*; *Brown et al., 2016*; *Loveland et al., 2016*). Interestingly, we observed that the NTD region of RelA is necessary and sufficient for NirD binding (*Figure 4B* and *Figure 5*). Moreover, our in vivo and in vitro results convincingly show that NirD can inhibit the (p)ppGpp synthesis from the constitutively active truncated RelA[N-terminal] (*Figure 6* and *Figure 4—figure supplement 1B*), supporting the idea of a ribosome-independent inhibition of the stringent response. Therefore, rather than preventing activation of RelA our results support a simple model of regulation in which NirD directly interacts with the catalytic domains of RelA to inhibit the synthetic activity. Such inhibition might be mediated by directly introducing structural changes in the catalytic domains. Alternatively, NirD may interact with RelA to avoid nucleotide binding by masking the nucleotide-binding site. Structural and functional analyses are currently under investigation to test whether direct binding of NirD inhibits RelA activity.

Finally, (p)ppGpp has emerged as an important regulator of not only the bacterial stress response but also of many aspects of bacterial physiology including virulence, immune evasion, and antibiotic tolerance. Therefore, strategies to inhibit RelA activity and the subsequent production of (p)ppGpp represent an attractive approach for the success of antimicrobial therapy. Currently, there are only a limited number of synthetic inhibitors known to target RelA, mainly based on the design of (p) ppGpp analogues (*Wexselblatt et al., 2012*; *Wexselblatt et al., 2013*). Hence, our identification that NirD constitutes a natural potent inhibitor of RelA in vivo and in vitro could represent an important resource for the future design of new drug compounds. Importantly, it is also likely to be a fascinating molecular tool to further tackle the functional and structural determinant of the stringent response.

## Materials and methods

### Experimental model and subject details

The *E. coli* strains and plasmids used in this study are listed in Appendix 1—key resources table. Cells were grown in lysogeny broth (LB) broth (*Miller, 1972*) with agitation or on NA (Thermo Fisher Scientific, MA). For cells containing plasmid(s), the media were appropriately supplemented with the antibiotic(s) kanamycin (25 µg/mL), chloramphenicol (50 µg/mL), and/or ampicillin (50 µg/mL).

### Plasmids construction

The plasmid derivatives were constructed by cloning into a plasmid a gene amplified by PCR from template DNA (chromosomal or plasmid) using the primers listed in Appendix 1—key resources table. The restriction enzymes as well as Shine-Dalgarno sequence for the pEG25 and pBbS2k plasmids are also indicated (Appendix 1—key resources table).

### Multicopy suppressor screen

The genetic assay for the identification of new regulator of (p)ppGpp homeostasis in *E. coli* was carried out as follows. A pool of plasmids from the ASKA library (*Kitagawa et al., 2005*) was introduced by electroporation into *E. coli* MG1655 cells containing the pBbS2k-*relA*. The transformed cells were then spread on NA plates containing the appropriate antibiotics and the inducers aTc (0 or 100 ng/mL) and IPTG (0, 100, or 200 µM) for induction of pBbS2k-*relA* and pCA24N derivatives from the ASKA library, respectively. After 36 hr of incubation at 37°C, clones that were able to form colonies under these non-permissive conditions were chosen for the plasmid preparation and sequencing. The pool of plasmids from the ASKA library was obtained by *Germain et al., 2019*.

## Growth assay

The ability of *E. coli* cells to grow under specific conditions was tested as follows. Single colonies were inoculated into 2 mL of LB broth, supplemented with the appropriate antibiotic(s), and cultured at 37˚C until stationary phase (~8 hr). The cultures were then serially diluted and spotted on NA, M9-glucose minimal medium, or SMG plates (*Supplementary file 1*) containing the appropriate antibiotic(s) and plasmid inducer(s). For growth assays on M9-glucose minimal medium or SMG, the cultures were washed twice and diluted in PBS before being serially diluted and spotted on agar plates. The cells were finally incubated at 37˚C overnight or ~36 hr depending on whether they were spotted on NA or minimal medium, respectively.

## In vivo (p)ppGpp measurement

The levels of (p)ppGpp in the cells were determined as described by *Germain et al., 2019*. Briefly, 100 µL of $^{32}$P-labeled cell samples were taken at set times and added to 40 µL of ice-cold 21 M formic acid to stop the reactions. The mixtures were then placed on ice for 20 min before being centrifuged at 4˚C for 20 min at 14,000 *g* to pellet cell debris and avoid them on chromatograms. 5 µL of each were loaded onto PEI-Cellulose thin layer chromatography (TLC) plates (Merck-Millipore) prior to ascending development with 1.5 M $KH_2PO_4$ solution (pH 3.4). Once fully developed, the TLC plates were dry at room temperature, separated from their upper part containing free $^{32}$P, and exposed overnight with a phosphor screen. The signals from the phosphor screen were then captured and quantified using an Amersham Typhoon Biomolecular Imager and ImageQuant TL 8.1 (GE Healthcare). We normalized the amount of (p)ppGpp to the total amount of G nucleotides observed in each sample, the total G being the sum of GTP, ppGpp, and pppGpp detected.

Regarding the preparation of the media, the labeling with $^{32}$P, and the induction of stress or starvation, they were carried out as follows. For overnight cultures, cells were grown in MOPS minimal medium (*Neidhardt et al., 1974*) supplemented with 0.2% glucose, 2 mM phosphate, and amino acids at 40 µg/mL (except for amino acid starvation experiment). Cultures were diluted 100 times in the same medium with 0.4 mM phosphate and incubated at 37˚C with shaking. At an $OD_{600nm}$ of ~0.5, they were diluted to an $OD_{600nm}$ of 0.05, labeled with 150 µCi of $^{32}$P, and grew to an $OD_{600nm}$ of ~0.20. At this point and 30 min later, inducers (100 ng/mL aTc or 1 mM IPTG) or 500 µg/mL L-valine were added for gene overexpression or induction of isoleucine starvation, respectively.

## Bacterial two-hybrid assay

In vivo protein-protein interactions were tested using the bacterial two-hybrid system (*Karimova et al., 1998*). Briefly, protein pairs were fused to the T18 and T25 domains of *B. pertussis* adenylate cyclase using the two compatible vectors pUT18C and pKT25, respectively. Co-transformed with the pairs of plasmids, single colonies of the *cya*-deficient strain BTH101 were inoculated into 2 mL of LB broth supplemented with ampicillin and kanamycin, and grown at 30˚C to stationary phase (~12 hr). 5 µL of the cultures were then spotted on NA plates containing the two antibiotics and 40 µg/mL X-gal as a color reporter for β-galactosidase and adenylate cyclase activities. The cells were finally incubated at 30˚C ~36 hr. To quantify the strength of the interaction, β-galactosidase activity was determined as described by *Miller, 1992* using stationary phase culture (~12 hr).

## Screening for loss of interaction by *nirD* random mutagenesis

*nirD* DNA coding sequence was mutagenized randomly through low-fidelity GoTaq polymerase amplification from pUT18C-*nirD* plasmid using 441 and 443 oligonucleotides (Appendix 1—key resources table). After 30 PCR cycles, amplified DNA was purified, diluted to 1:100,000 and further amplified for 30 cycles. Purified DNA were cloned into pUT18C plasmid. BTH101 cells harboring pKT25-*relA* were then transformed with the resulting pUT18C-*nirD* mutated library and screened on X-Gal plates. White colonies were selected and sequenced.

## MG1655 *nirD*$^{E50K}$ construction

The chromosomal mutant strain *nirD*$^{E50K}$ was constructed by scarless mutagenesis using a two-step lambda Red system adapted from *Blank et al., 2011*. DNA fragments containing a chloramphenicol resistance marker and the meganuclease I-*SceI* recognition site were amplified by PCR using primers 629 and 630 with pWRG100 as template. The resulting PCR product was electroporated into

MG1655 cells, which had induced lambda recombinase expression for 1 hr from plasmid pKD46 (*Datsenko and Wanner, 2000*). After 1 hr of phenotypic expression, the cells were plated on NA plates containing 25 µg/mL chloramphenicol. Selected colonies contained the *nirD::cat* I-*Sce*I allele, which was subsequently transduced in MG1655 cells harboring the pWRG99 plasmid (*Blank et al., 2011*). The mutant allele *nirD$^{E50K}$* was amplified from pEG25-*nirD$^{E50K}$* using primers 655 and 656. The PCR product was then electroporated into MG1655 *nirD::cat* I-*Sce*I expressing lambda recombinase from pWRG99 and after 1 hr of phenotypic expression, cells were serially diluted and plated on NA plates containing 100 µg/mL ampicillin and 1 µg/mL aTc. pWRG99 also encoded for the meganuclease I-*Sce*I under the control of an aTc-inducible promoter. The proper integration of the *nirD$^{E50K}$* was confirmed by diagnostic PCR and sequencing.

## Growth curves

Growth of WT cells, Δ*nirD,* and *nirD$^{E50K}$* mutants was measured under anaerobiosis as follows. Stationary-phase cultures were washed in PBS, diluted to an $OD_{600nm}$ of ~0.03 in M9-glucose minimal medium without amino acids or supplement with 1 mM SMG in Hungate tubes, and air was evacuated and replaced with $N_2$. The cultures were then incubated at 37°C and growth was followed by measurement of $OD_{600nm}$. Although at least three independent experiments were performed, representative growth curves are shown. Indeed, day-to-day variations in medium composition shifted the curves slightly, but the sample-to-sample comparisons were invariable.

For the functional test of the NADH-dependent nitrite reductase activity, the cultures were prepared as described above except that they were diluted in a defined medium supplemented with 140 mM glycerol and 100 mM nitrate. The defined medium is composed of $Na_2HPO_4$ (60 mM), $KH_2PO_4$ (22 mM), NaCl (8 mM), $MgSO_4$ (1 mM), $CaCl_2$ (100 µM), thiamine (1 µg/mL), sodium molybdate (5 µM), sodium selenite (1 µM), and $FeSO_4$ (10 µM).

## Protein expression and production

BL21 (DE3)-competent cells were transformed with the pEG25-*relA$^{N-terminal}$*, pET28a(+)-*nirD* or pET28a(+)-*nirD$^{E50K}$* and plated on selective NA. For RelA$^{N-terminal}$ production, several colonies were picked up and inoculated into 100 mL LB containing 100 µg/mL ampicillin. The cultures were grown with shaking at 30°C overnight. Overnight cultures were dispersed as inoculum of 20 mL aliquots into 1 L Terrific Broth media with 100 µg/mL ampicillin and shaken at 30°C until $OD_{600nm}$ ~0.5, then cells were induced with 0.5 mM IPTG and incubated at 30°C for 4 hr. For NirD and NirD$^{E50K}$ production, several colonies were picked up and inoculated into 100 mL LB containing 50 µg/mL kanamycin. The cultures were grown with shaking at 30°C overnight. Overnight cultures were dispersed as inoculum of 20 mL aliquots into 1 L LB media with 50 µg/mL kanamycin and shaken at 30°C until $OD_{600nm}$ ~0.5, then cells were induced with 0.5 mM IPTG and incubated at 30°C for 4 hr. Finally, cells were centrifuged at 9000 *g* for 20 min at 4°C. Dry cell pellet was stored at −80°C.

## Protein purification

For NirD and NirD$^{E50K}$ purification, cells were suspended in lysis buffer (50 mM Tris-HCl [pH 8.0], 300 mM NaCl, 1 mM EDTA, 0.5 mg/mL lysozyme, 1 mM phenylmethylsulfonyl fluoride [PMSF], 20 µg/mL DNase, and 20 mM $MgCl_2$), incubated for 1 hr at 4°C with gentle shaking, and then subjected to three cycles of French-press lysis steps. The soluble fraction was obtained by centrifugation for 20 min at 200,000 *g*. Recombinant proteins were purified by ion metal affinity chromatography using a 5 mL nickel (HiTrap$^{HP}$) column on an ÄKTA pure 25 (GE Healthcare) pre-equilibrated in 50 mM Tris-HCl (pH 8.0), 300 mM NaCl, and 10 mM imidazole (buffer A). After several washes in buffer A, His-tagged proteins were eluted in buffer A supplemented with 250 mM imidazole and directly desalted using HiPrep 26/10 Desalting column pre-equilibrated with buffer A. The desalted proteins were mixed with 0.2 mg/mL of TEV protease, incubated for 2 hr at 25°C, and then loaded onto a HisTrap column pre-equilibrated in buffer A to retain TEV, TRX, uncleaved proteins, and contaminants. Untagged NirD or NirD$^{E50K}$ were collected in the flow-through, concentrated on a Centricon (Millipore; cutoff of 3 kDa), and passed through a HiLoad 16/600 Superdex 200 column pre-equilibrated with 50 mM Tris-HCl (pH 8.0), 300 mM NaCl, 2 mM β-mercaptoethanol, and 2% glycerol. The purity of NirD preparations was assessed by SDS-PAGE (*Figure 5—figure supplement 1B, C*).

For RelA$^{N-terminal}$ purification, cells were suspended in lysis buffer (50 mM Tris-HCl [pH 8.0], 500 mM NaCl, 10 mM imidazole, 2 mM β-mercaptoethanol, 0.5% CHAPS, 2% glycerol, 1 mM EDTA, 0.5 mg/mL lysozyme, 1 mM PMSF, 20 μg/mL DNase, and 20 mM MgCl$_2$), incubated for 1 hr at 4°C with gentle shaking, and then subjected to three cycles of French-press lysis steps. The soluble fraction was obtained by centrifugation for 20 min at 200,000 g. Recombinant proteins were purified by ion metal affinity chromatography using a 5 mL nickel (HiTrap$^{HP}$) column on an ÄKTA pure 25 (GE Healthcare) pre-equilibrated with equilibrium buffer (50 mM Tris-HCl [pH 8.0], 500 mM NaCl, 10 mM imidazole, 2 mM β-mercaptoethanol, and 2% glycerol), eluted in elution buffer (50 mM Tris-HCl [pH 8.0], 250 mM NaCl, 500 mM imidazole, 2 mM β-mercaptoethanol, and 2% glycerol), and directly desalted using HiPrep 26/10 Desalting column pre-equilibrated with buffer containing 50 mM Tris-HCl (pH 8.0), 500 mM NaCl, 500 mM KCl, 2 mM β-mercaptoethanol, and 2% glycerol. Desalted proteins were concentrated on a Centricon (Millipore; cutoff of 10 kDa) and then subjected to SEC purification using a HiLoad 16/600 Superdex 200 column pre-equilibrated with 50 mM Tris-HCl (pH 8.0), 500 mM NaCl, 500 mM KCl, 2 mM β-mercaptoethanol, and 2% glycerol. The purity of RelA preparations was assessed by SDS-PAGE (*Figure 5—figure supplement 1A*).

## Protein-protein interaction analysis by SEC

Purified RelA$^{N-terminal}$ and NirD were either alone or pre-mixed in a 1:1 ratio (10 μM each) in buffer containing 50 mM Tris-HCl (pH 8.0), 500 mM NaCl, 500 mM KCl, 2 mM β-mercaptoethanol, and 2% glycerol prior separation on a Superdex 200 10/300 GL using an ÄKTA pure 25 apparatus (GE Healthcare) pre-equilibrated with the same buffer. Elution fractions were then analyzed with SDS-PAGE.

## Biolayer interferometry

NirD or NirD$^{E50K}$ were biotinylated using the EZ-Link NHS-PEG4-Biotin (Thermo Scientific) at 4°C for 2 hr. The reaction was stopped by removing the excess of the biotin using a Zeba Spin Desalting column (Thermo Scientific). BLI studies were performed at 25°C using the Blitz apparatus (FortéBio) with shaking at 2200 rpm with the following steps: 30 s baseline, 120 s association, and 120 s dissociation. Streptavidin biosensor tips (FortéBio) were hydrated with 0.350 mL of 50 mM Tris-HCl (pH 8.0), 500 mM NaCl, 500 mM KCl, 2 mM β-mercaptoethanol, and 2% glycerol for 10 min and then loaded with 3 μM of biotinylated NirD or NirD$^{E50K}$ in the same buffer. The biosensors were then incubated with 10 μg/mL biocytin in interaction buffer (50 mM Tris-HCl [pH 8.0], 500 mM NaCl, 500 mM KCl, 2 mM β-mercaptoethanol, 2% glycerol, and 1 mg/mL) for 90 s to avoid the non-specific binding RelA$^{N-terminal}$ to the streptavidin biosensors. To study the binding of NirD or NirD$^{E50K}$ to RelA$^{N-terminal}$, increasing concentrations RelA$^{N-terminal}$ (1–30 μM) in interaction buffer were used and the association and dissociation phases were monitored, respectively. In all experiments, the BLItz Pro Software performed a reference subtraction of the RelA$^{N-terminal}$ protein response on the uncoated biosensors for each tested concentration. The dissociation constant ($K_D$), that is affinity of NirD to RelA$^{N-terminal}$, was estimated using the GraphPad Prism 5.0 software on the basis of the steady-state-level responses in nm at equilibrium. The $K_D$ was estimated by plotting on x axis the different concentrations and the different responses of RelA$^{N-terminal}$ at the saturation (10 s before the end of the association) on the y axis. For $K_D$ calculation, a nonlinear regression fit for xy analysis was used and one site (specific binding) as a model [corresponding to the equation $y=Bmax*x/(K_D+x)$].

## Isothermal titration calorimetry

ITC was performed to determine the affinity between NirD and RelA$^{N-terminal}$. In a typical setup, RelA$^{N-terminal}$ (30 μM) in buffer containing 50 mM Tris-HCl (pH 8.0), 500 mM NaCl, 500 mM KCl, 2 mM β-mercaptoethanol, and 2% glycerol was placed in the cell, and NirD (or NirD$^{E50K}$) ligand (300 μM in the same buffer) was placed in the titration syringe. All experiments were carried out at 25°C with a stirring speed of 750 rpm and a reference power of 5 μcal/s using the MicroCal PEAQ-ITC (Malvern Panalytical) with 19 injections (0.4 μL for the initial injection and 2 μL for the next 18). Each injection lasted for 4 s with a space of 150 s between each injection. ITC measures the energy released or absorbed by a chemical reaction, and the energy of the interaction is proportional to the amount of ligand that binds to the protein. These heat pulses are recorded for each injection and then integrated with respect to time and are normalized to the concentration of the reaction to

generate kcal/mol versus the molar ratio (ligand/sample). The enthalpy of the reaction is obtained, and with a 'One Set of Sites' model in the PEAQ-ITC Analysis Software, the $K_D$ and stoichiometry is determined. The fitted offset option in the PEAQ-ITC Analysis Software was used to correct for the heat of the dilution.

### pppGpp synthesis assay

The pppGpp synthetic activity of RelA[N-terminal] was assayed at 37°C in a buffer containing 50 mM Tris-HCl (pH 8.0), 300 mM NaCl, 2 mM β-mercaptoethanol, 2% glycerol, and 15 mM $Mg^{2+}$. Typically, reaction mixtures contained 8 mM ATP, 6 mM GTP and [$\alpha$-[32]P]GTP, 1 µM RelA[N-terminal] with or without NirD (or NirD[E50K]) at concentrations ranging from 0.5 to 16 µM, and reactions were started by adding pre-warmed nucleotides to the protein(s). During the time course of the reaction, 20 µL samples were taken and added to 8 µL of ice-cold 21 M formic acid to stop the reactions. The time points were resolved by TLC, and radio-labeled nucleotides were quantified as described in the 'In vivo (p)ppGpp measurement' section.

## Acknowledgements

This work was supported by the European Research Council starting grant (ERC StG) under the European Union's Horizon 2020 and innovation program grant agreement no. 714934 'Stringency' to EM. We thank Farida Seduk, Aurore Jacq-Bailly, Gaël Brasseur, Axel Magalon, and Benjamin Ezraty for advices and stimulating discussions for anaerobic experiments.

## Additional information

### Funding

| Funder | Grant reference number | Author |
| --- | --- | --- |
| H2020 European Research Council | 714934 | Etienne Maisonneuve |

The funders had no role in study design, data collection and interpretation, or the decision to submit the work for publication.

### Author contributions

Loïc Léger, Conceptualization, Data curation, Formal analysis, Validation, Investigation, Visualization, Methodology, Writing - original draft, Writing - review and editing; Deborah Byrne, Paul Guiraud, Data curation, Formal analysis, Methodology, Writing - review and editing; Elsa Germain, Conceptualization, Supervision, Validation, Methodology, Project administration, Writing - review and editing; Etienne Maisonneuve, Conceptualization, Data curation, Supervision, Funding acquisition, Validation, Investigation, Writing - original draft, Project administration, Writing - review and editing

### Author ORCIDs

Loïc Léger https://orcid.org/0000-0002-3641-3697
Etienne Maisonneuve https://orcid.org/0000-0003-3451-1223

### Decision letter and Author response

Decision letter https://doi.org/10.7554/eLife.64092.sa1
Author response https://doi.org/10.7554/eLife.64092.sa2

## Additional files

### Supplementary files

- Supplementary file 1. Media used in this work.
- Transparent reporting form

## Data availability

All data generated or analysed during this study are included in the manuscript and supporting files. Source data files have been provided for Figure 1, Figure 2, Figure 3, Figure 4, Figure 5, Figure 6, Figure 3-figure supplement 1, Figure 4-figure supplement 1 and Figure 5-figure supplement 1.

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

# Appendix 1

**Appendix 1—key resources table**

| Reagent type (species) or resource | Designation | Source or reference | Identifiers | Additional information |
|---|---|---|---|---|
| strain, strain background (*Escherichia coli*) | BTH101 | *Karimova et al., 1998* doi: 10.1073/pnas.95.10.5752 | | F⁻ *cya-99 araD139 galE15 galK16 rpsL* (*Str^R*) *hsdR2 mcrA1 mcrB1 relA1* |
| strain, strain background (*E. coli*) | BL21 (DE3) | New England Biolabs | | B F⁻ *ompT gal* |
| strain, strain background (*E. coli*) | MG1655; WT | Lab collection | | Wild-type |
| strain, strain background (*E. coli*) | *nirD^E50K* | This paper | | MG1655 *nirD^E50K* |
| strain, strain background (*E. coli*) | Δ*nirD* | This paper | | MG1655 *nirD::FRT*; P1 from KEIO collection (resistance cassette has been flipped out) (*Baba et al., 2006*) |
| strain, strain background (*E. coli*) | Δ*relA* | *Germain et al., 2019* doi: 10.1038/s41467-019-13764-4 | | MG1655 *relA::FRT* |
| strain, strain background (*E. coli*) | Δ*relA spoT* | This paper | | MG1655 relA::FRT spoT207::cat; P1 transduction from CF1693 (*Baba et al., 2006*) in MG1655 *relA::FRT* |
| recombinant DNA reagent | pBbS2k-*rfp* | *Lee et al., 2011* doi: 10.1186/1754-1611-5-12 | | Kan^R, P_{tet}, *rfp* |
| recombinant DNA reagent | pBbS2k-*relA* | This paper | | Kan^R, P_{tet}, *relA*; primers 16/17; Maisonneuve lab |
| recombinant DNA reagent | pBbS2k-*relA* (sd4) | This paper | | Kan^R, P_{tet}, sd4, *relA*; primers 163/17; Maisonneuve lab |
| recombinant DNA reagent | pBbS2k-*relA^{N-terminal}* | This paper | | Kan^R, P_{tet}, *relA^{N-terminal}*; primers 16/557; Maisonneuve lab |
| recombinant DNA reagent | pEG25 | *Germain et al., 2019* doi: 10.1038/s41467-019-13764-4 | | Amp^R, P_{T5-lac} |
| recombinant DNA reagent | pEG25-*nirD* | This paper | | Amp^R, P_{T5-lac}, *nirD*; primers 386/387; Maisonneuve lab |
| recombinant DNA reagent | pEG25-*spoT* (sd8U) | *Germain et al., 2019* doi: 10.1038/s41467-019-13764-4 | | Amp^R, P_{T5-lac}, sd8U, *spoT* |
| recombinant DNA reagent | pEG25-*nirD^E50K* | This paper | | Amp^R, P_{T5-lac}, *nirD^E50K*; primers 386/387; Maisonneuve lab |
| recombinant DNA reagent | pEG25-*nirD* (WTsd) | This paper | | Amp^R, P_{T5-lac}, WTsd, *nirD*; primers 555/387; Maisonneuve lab |

*Continued on next page*

*Appendix 1—key resources table continued*

| Reagent type (species) or resource | Designation | Source or reference | Identifiers | Additional information |
|---|---|---|---|---|
| recombinant DNA reagent | pEG25-*nirBD* (WTsd) | This paper | | Amp$^R$, P$_{T5-lac}$, WTsd, *nirBD*; primers 553/544; Maisonneuve lab |
| recombinant DNA reagent | pEG25-*relA*$^{N-terminal}$ | *Germain et al., 2019* doi: [10.1038/s41467-019-13764-4](10.1038/s41467-019-13764-4) | | Amp$^R$, P$_{T5-lac}$, *relA*$^{N-terminal}$ |
| recombinant DNA reagent | pKD46 | *Datsenko and Wanner, 2000* doi: [10.1073/pnas.120163297](10.1073/pnas.120163297) | | Amp$^R$, P$_{ara}$, λ Red |
| recombinant DNA reagent | pKT25 | *Karimova et al., 1998* doi: [10.1073/pnas.95.10.5752](10.1073/pnas.95.10.5752) | | Kan$^R$ |
| recombinant DNA reagent | pKT25-*zip* | *Karimova et al., 1998* doi: [10.1073/pnas.95.10.5752](10.1073/pnas.95.10.5752) | | Kan$^R$, *zip* |
| recombinant DNA reagent | pKT25-*relA* | *Germain et al., 2019* doi: [10.1038/s41467-019-13764-4](10.1038/s41467-019-13764-4) | | Kan$^R$, *relA* |
| recombinant DNA reagent | pKT25-*spoT* | *Germain et al., 2019* doi: [10.1038/s41467-019-13764-4](10.1038/s41467-019-13764-4) | | Kan$^R$, *spoT* |
| recombinant DNA reagent | pKT25-*nirB* | This paper | | Kan$^R$, *nirB*; primers 617/618; Maisonneuve lab |
| recombinant DNA reagent | pKT25-*relA*$^{1-663}$ | This paper | | Kan$^R$, *relA*$^{1-663}$; primers 497/502; Maisonneuve lab |
| recombinant DNA reagent | pKT25-*relA*$^{1-580}$ | This paper | | Kan$^R$, *relA*$^{1-580}$; primers 497/501; Maisonneuve lab |
| recombinant DNA reagent | pKT25-*relA*$^{1-470}$ | This paper | | Kan$^R$, *relA*$^{1-470}$; primers 497/500; Maisonneuve lab |
| recombinant DNA reagent | pKT25-*relA*$^{N-terminal}$ | This paper | | Kan$^R$, *relA*$^{N-terminal}$; primers 497/499; Maisonneuve lab |
| recombinant DNA reagent | pKT25-*relA*$^{1-181}$ | This paper | | Kan$^R$, *relA*$^{1-181}$; primers 497/498; Maisonneuve lab |
| recombinant DNA reagent | pKT25-*relA*$^{181-744}$ | This paper | | Kan$^R$, *relA*$^{181-744}$; primers 531/533; Maisonneuve lab |
| recombinant DNA reagent | pKT25-*spoT*$^{N-terminal}$ | *Germain et al., 2019* doi: [10.1038/s41467-019-13764-4](10.1038/s41467-019-13764-4) | | Kan$^R$, *relA*$^{N-terminal}$ |
| recombinant DNA reagent | pUT18C | *Karimova et al., 1998* doi: [10.1073/pnas.95.10.5752](10.1073/pnas.95.10.5752) | | Amp$^R$ |
| recombinant DNA reagent | pUT18C-*zip* | *Karimova et al., 1998* doi: [10.1073/pnas.95.10.5752](10.1073/pnas.95.10.5752) | | Amp$^R$, *zip* |
| recombinant DNA reagent | pUT18C-*nirD* | This paper | | Amp$^R$, *nirD*; primers 441/443; Maisonneuve lab |

*Continued on next page*

*Appendix 1—key resources table continued*

| Reagent type (species) or resource | Designation | Source or reference | Identifiers | Additional information |
|---|---|---|---|---|
| recombinant DNA reagent | pUT18C-*nirD*$^{E50K}$ | This paper | | Amp$^R$, *nirD*$^{E50K}$; primers 441/443; Maisonneuve lab |
| recombinant DNA reagent | pET28a(+) | *Germain et al., 2019* doi: 10.1038/s41467-019-13764-4 | | Kan$^R$, Trx, His |
| recombinant DNA reagent | pET28a(+)-*nirD* | This paper | | Kan$^R$, Trx, His, TEV, *nirD*; primers 558/544; Maisonneuve lab |
| recombinant DNA reagent | pET28a(+)-*nirD*$^{E50K}$ | This paper | | Kan$^R$, Trx, His, TEV, *nirD*$^{E50K}$; primers 558/544; Maisonneuve lab |
| recombinant DNA reagent | pWRG99 | *Blank et al., 2011* doi: 10.1371/journal.pone.0015763 | | pKD46 with Ptet, I-SceI |
| recombinant DNA reagent | pWRG100 | *Karimova et al., 1998* doi: 10.1073/pnas.95.10.5752 doi: 10.1371/journal.pone.0015763 | | pKD3 with I-*Sce*I recognition site |
| sequence-based reagent | Primer 16 | This paper | | CCCCGAATTCGTCGAC TCAAGGAGGT TTTATAAATGGTTGCGGTAAGAAG TGCA; restriction enzyme EcoRI; Maisonneuve lab |
| sequence-based reagent | Primer 17 | This paper | | CCCCGGATCCCTAACTCCCG TGCAACCGAC; restriction enzyme BamHI; Maisonneuve lab |
| sequence-based reagent | Primer 163 | This paper | | CCCCGAATTCGTCGACTCAAGGAT TAAATGGTTGCGGTAAGAAGTGCA; restriction enzyme EcoRI; Maisonneuve lab |
| sequence-based reagent | Primer 386 | This paper | | CCCCGAATTCGTCGACTCAAGGAG GTTTTATAAATGAGCCAG TGGAAAGACAT; restriction enzyme EcoRI; Maisonneuve lab |
| sequence-based reagent | Primer 387 | This paper | | CCCCGGATCCTTAACCG CGCAGCTGCACCA; restriction enzyme BamHI; Maisonneuve lab |
| sequence-based reagent | Primer 441 | This paper | | CCCCTCTAGAAATGAGCC AGTGGAAAGACAT; restriction enzyme XbaI; Maisonneuve lab |
| sequence-based reagent | Primer 443 | This paper | | CCCCGGTACCTTAACC GCGCAGCTGCACCA; restriction enzyme KpnI; Maisonneuve lab |
| sequence-based reagent | Primer 497 | This paper | | CCCCTCTAGAAATGGTT GCGGTAAGAAGTGC; restriction enzyme XbaI; Maisonneuve lab |
| sequence-based reagent | Primer 498 | This paper | | CCCCGGTACCGATGTTGGTACAC TCTTTTG; restriction enzyme KpnI; Maisonneuve lab |
| sequence-based reagent | Primer 499 | This paper | | CCCCGGTACCTTCGTCGAGCA TTTCGCCGG; restriction enzyme KpnI; Maisonneuve lab |

*Continued on next page*

*Appendix 1—key resources table continued*

| Reagent type (species) or resource | Designation | Source or reference | Identifiers | Additional information |
|---|---|---|---|---|
| sequence-based reagent | Primer 500 | This paper | | CCCCGGTACCGTTCGGCTGTTTC TGGGTGA; restriction enzyme KpnI; Maisonneuve lab |
| sequence-based reagent | Primer 501 | This paper | | CCCCGGTACCAAGTTGC TTCAGCGCGGCGG; restriction enzyme KpnI; Maisonneuve lab |
| sequence-based reagent | Primer 502 | This paper | | CCCCGGTACCGGCGGAGTAGCTC TCACCCC; restriction enzyme KpnI; Maisonneuve lab |
| sequence-based reagent | Primer 531 | This paper | | CCCCTCTAGAATACGCACCGC TGGCTAACCG; restriction enzyme XbaI; Maisonneuve lab |
| sequence-based reagent | Primer 533 | This paper | | CCCCGGTACCCTAACTCCCG TGCAACCGAC; restriction enzyme KpnI; Maisonneuve lab |
| sequence-based reagent | Primer 544 | This paper | | CCCCAAGCTTTTAACCGCGCAGC TGCACCA; restriction enzyme HindIII; Maisonneuve lab |
| sequence-based reagent | Primer 553 | This paper | | CCCCGAATTCAATAGAAAAGAAA TCG AGGCAAAAATGAGCAAAG TCAGACTCGC; restriction enzyme EcoRI; Maisonneuve lab |
| sequence-based reagent | Primer 555 | This paper | | CCCCGAATTCAGTAACTCTGG TGGAGG ACAACGCATGAGCCAG TGGAAAGACAT; restriction enzyme EcoRI; Maisonneuve lab |
| sequence-based reagent | Primer 557 | This paper | | CCCCGGATCCTTATTCGTCGAGCA TTTCGCCGG; restriction enzyme BamHI; Maisonneuve lab |
| sequence-based reagent | Primer 558 | This paper | | CCCCGGATCCGAGAACCTGTACTT CCAATCAATGAGCCAG TGGAAAGACAT; restriction enzyme BamHI; Maisonneuve lab |
| sequence-based reagent | Primer 617 | This paper | | CCCCTCTAGAAATGAGCAAAG TCAGACTCGC; restriction enzyme XbaI; Maisonneuve lab |
| sequence-based reagent | Primer 618 | This paper | | CCCCGGTACCTCATGCGTTGTCC TCCACCA; restriction enzyme KpnI; Maisonneuve lab |
| sequence-based reagent | Primer 629 | This paper | | ATGAGCCAGTGGAAAGACATCTG CAAAATCGATGACATCCTGCCTGA AACCGCCTTACGCCCCGCCCTGC; Maisonneuve lab |
| sequence-based reagent | Primer 630 | This paper | | TTAACCGCGCAGCTGCACCACGC CGTCTTTCACTCGCGCTTCGTAAT GTTCTAGACTATATTACCCTGTT; Maisonneuve lab |
| sequence-based reagent | Primer 655 | This paper | | ATGAGCCAGTGGAAAGACAT; Maisonneuve lab |
| sequence-based reagent | Primer 656 | This paper | | TTAACCGCGCAGCTGCACCA; Maisonneuve lab |
| chemical compound, drug | [$\alpha^{32}$P]GTP | PerkinElmer | Catalog number: BLU006H250UC | |

*Continued on next page*

*Appendix 1—key resources table continued*

| Reagent type (species) or resource | Designation | Source or reference | Identifiers | Additional information |
|---|---|---|---|---|
| chemical compound, drug | $^{32}$P | PerkinElmer | Catalog number: NEX053H001MC | |
| chemical compound, drug | EZ-Link NHS-PEG4-Biotin | Thermo Scientific | Catalog number: 21330 | |
| software, algorithm | BLItz Pro Software | | https://www.sartorius.com/en/products/protein-analysis | |
| software, algorithm | MicroCal PEAQ-ITC Analysis Software | Malvern Panalytical | https://www.malvernpanalytical.com/fr/products/product-range/microcal-range/microcal-itc-range/microcal-peaq-itc | |
| software, algorithm | Prism 5.0 software | GraphPad | https://www.graphpad.com/scientific-software/prism/ | |
| other | HiLoad 16/600 Superdex 200 column | GE Healthcare | Catalog number: GE28-9893-35 | |
| other | HiPrep 26/10 Desalting column | GE Healthcare | Catalog number: GE17-5087-01 | |
| other | HisTrap column | GE Healthcare | Catalog number: GE17-5248-02 | |
| other | Streptavidin Biosensors | FortéBio | Catalog number: 18-5117 | |
| other | Superdex 200 10/300 GL | GE Healthcare | Catalog number: GE28-9909-44 | |
| other | Zeba Spin Desalting column | Thermo Scientific | Catalog number: 89882 | |

