## [Decision Letter]

**Acceptance summary:**

The reviewers all agreed that the work is very interesting and intriguing and potentially of great interest as you report the identification of NirD, the small subunit of a nitrite reductase, as a new negative regulator of the stringent response in *E. coli*. Seeing the importance of the stringent response in bacterial adaptation to environmental stresses, it is potentially a significant advance.

**Decision letter after peer review:**

Thank you for submitting your article "NirD Curtails the Stringent Response by Inhibiting RelA Activity in *Escherichia coli*" for consideration by *eLife*. Your article has been reviewed by 4 peer reviewers, and the evaluation has been overseen by a Reviewing Editor and Gisela Storz as the Senior Editor. The following individual involved in review of your submission has agreed to reveal their identity: Severin Ronneau (Reviewer #1).

The reviewers have discussed the reviews with one another and the Reviewing Editor has drafted this decision to help you prepare a revised submission.

As the editors have judged that your manuscript is of interest, but as described below that additional experiments are required before it is published, we would like to draw your attention to changes in our revision policy that we have made in response to COVID-19 (https://elifesciences.org/articles/57162). First, because many researchers have temporarily lost access to the labs, we will give authors as much time as they need to submit revised manuscripts. We are also offering, if you choose, to post the manuscript to bioRxiv (if it is not already there) along with this decision letter and a formal designation that the manuscript is "in revision at eLife". Please let us know if you would like to pursue this option. (If your work is more suitable for medRxiv, you will need to post the preprint yourself, as the mechanisms for us to do so are still in development.)

The reviewers all agreed that the work is very interesting and intriguing and potentially of great interest as you report through a multicopy suppressor screen the identification of NirD, the small subunit of a nitrite reductase, as a new negative regulator of the stringent response in *E. coli*. Seeing the importance of the stringent response in bacterial adaptation to environmental stresses, it is potentially a significant advance. Specifically, overexpression of NirD very convincingly inhibits RelA-dependent production of (p)ppGpp, NirD interacts with the N-terminal domain of RelA, both in vitro and in *E. coli*, leading to reduction of (p)ppGpp synthesis.

However, all reviewers were in agreement that additional work is needed to establish the physiological relevance of this regulation. Although solid and well controlled, all the experiments rely on overexpression of NirD. Reviewers reckon that evidence of the importance of the interaction between NirD and RelA through phenotypic and molecular characterizations of null and point mutants is required before we can accept the manuscript for publication. They have agreed on a minimal set of experiments that are essential if you chose to revise the manuscript. In addition, they were also all of the opinion that the text should be revised to reflect the most up-to-date and relevant literature, and add depth to the discussion. Their main requests are the following:

Essential revisions:

– Measure levels of (p)ppGpp in wild-type *E. coli*, nirD, nirD_E50K, nirB and nirBD mutants in aa starvation, stationary phase, aerobic and anaerobic growth.

– Report the effects of nirD or nirBD mutations on growth under the above environmental conditions.

– Document the ratio of RelA:NirD. Is RelA in a much higher abundance than NirD? How does this relate to the importance of the interaction?

– Revise the text according to the following remarks:

a – Page 3, lines 12-17: with reference to the targets and roles of (p)ppGpp, the authors cite Atkinson et al., 2011; Kanjee et al. 2012; Hauryliuk et al. 2015 and Potrykus and Cashel 2008.

Remove here the reference of Atkinson et al., 2011, which described the phylogenic distribution of RSH enzymes.

– Add to or replace Kanjee et al. 2012 more recent papers in which they identified (p)ppGpp targets: Corrigan et al., 2016 (PMID: 26951678); Zhang et al., 2018 (PMID: 29511080); Wang et al., 2019 (PMID: 30559427)

– Add to or replace Hauryliuk et al. 2015 and Potrykus and Cashel 2008 by updated reviews: Hengge 2020 (PMID: 32244175); Irving et al. 2020 (PMID: 33149273)

b – Page 3, Line 23: in relation to the CTD, references Pausch 2020 Cell Reports, Takada 2020 Front Microbiol, and Gratani 2018 Plos Genetics should be included.

c – When describing RelA and SpoT as well as the mechanisms regulating their activities, the authors refer to only two research papers (Hogg et al., 2004; Mechold et al., 2002) and again the same reviews (Hauryliuk et al., 2015; Potrykus and Cashel 2008). Add Tamman et al., 2020 (PMID: 32393900)

d – Page 4, Line 4: two recent reviews extensively describe how stress signals are sensed by RSH enzymes. In place of Potrykus and Cashel, 2008, cite these:

Irving SE, Corrigan RM. (PMID: 29493495); Ronneau S, Hallez R. (PMID: 30980074)

e – Page 4, line 24, remove the "in" after Rsd.

f – Page 4: It would be interesting to add that RelA can also be activated by other stressful conditions than aa starvation which ultimately leads to aa starvation, e.g. fatty acid starvation that leads to lysine starvation (Sinha et al., 2019, PMID: 31400173).

g – Comment on why GppA relieves toxicity? pppGpp is less toxic than ppGpp, why?

h – Page 8, line 20 and page 13, line 9: add reports describing RSH regulation by

– the competence proteins ComGA: Hahn et al., 2015 (PMC4722805)

– the phosphotransferase system: Karstens et al., 2014 (PMID: 24515609); Ronneau et al., 2016 (PMID: 27109061); Ronneau et al., 2019 (PMID: 30496454).

i – Page 8, line 3, "zing" correct the typo.

j – Page 11, line7 – change to S5B and C to S4B and C. Also include A-E labels in figure S4 and a specific legend for S4E.

k – Page 19 line 15 – change broh to broth.

l – What do you mean by "ribosome-independent inhibition of the stringent response"?

---

## [Author Response]

Essential revisions:– Measure levels of (p)ppGpp in wild-type E. coli, nirD, nirD_E50K, nirB and nirBD mutants in aa starvation, stationary phase, aerobic and anaerobic growth.

We measured (p)ppGpp level of wild-type (WT), ΔnirD, and ΔrelA mutants during aerobic amino acid starvation and stationary phase entry (when (p)ppGpp rised). As shown, no significant difference is observed between WT and ΔnirD. These results are not unexpected given that nirD is not induced under these growth conditions. Indeed, nirD is induced during anaerobic growth (Harborne et al., 1992; Peakman et al., 1990) and its expression is maximized in the presence of glucose and nitrate (Tyson et al., 1997; Wang and Gunsalus, 2000).

For amino acid starvation (Author response image 1, left panel), exponentially growing cells of WT, ΔnirD, and ΔrelA were challenged by addition of 500 μg/mL of L-valine. Nucleotides were extracted from samples collected at the indicated times and separated by TLC. The autoradiogram is representative of 3 independent experiments. For measurement of stationary phase (p)ppGpp levels (Author response image 1, right panel), exponentially growing cells of WT, ΔnirD, and ΔrelA were labeled at OD600nm ~0.25 and samples were collected from 3 hours later, at the entrance into stationary phase. Nucleotides were extracted and separated by TLC. The autoradiogram is representative of 3 independent experiments.

**Author response image 1. sa2fig1:** *nirD* does not seem to impact accumulation of (p)ppGpp during amino acid starvation or in stationary phase under aerobic conditions.

Unfortunately, with respect to safety rules, we were technically not able to address this question during anaerobic growth conditions. Indeed, rules regarding the use of radioelements are strictly controlled and experiments have to be performed in a restricted area which is not equipped with an anaerobic glove box. Moreover, experiments using hungate tubes are not tolerated since tubes are under pressure and required needles for sampling and finally would also require massive and unwarranted use of radioelements.

– Report the effects of nirD or nirBD mutations on growth under the above environmental conditions.

We first performed growth experiments in aerobic conditions and as shown, no significant difference is observed between WT and Δ*nirD* mutant. As explained above, *nirD* is maximally expressed in anaerobic growth conditions in presence of glucose and nitrate. Note also that the *relA* mutant does not display a significant phenotype under these conditions.

Growth curves of WT cells, ΔnirD and ΔrelA mutants under aerobic amino acid starvation (Author response image 2, left panel) and stationary phase (Author response image 2, right panel). Stationary-phase cultures were diluted 100 times in MOPS-glucose minimal medium. The cultures were then incubated at 37°C and growth was followed by measurement of OD600nm using microplate reader. To induce amino acid starvation, 500 μg/mL of L-valine was added at the indicated time. Data are represented as the means of 3 independent experiments, error bars depict the SDs.

**Author response image 2. sa2fig2:** *nirD* does not seem to affect growth during amino acid starvation or in stationary phase under aerobic conditions.

As mentioned above, nirD is induced during anaerobic growth and its expression is maximized in the presence of glucose and nitrate. Moreover, the cytoplasmic nitrite reductase NirBD is a key regulator of the nitric oxide level during nitrate respiration and as a result a strain devoid of nitrite reductase activity presents a severe growth defect under nitrate respiration (Bulot et al., 2019). This led us to the conclusion that these conditions were not appropriated to dissociate the importance of NirD in the nitrite reductase activity to that of the regulation of RelA activity.

Therefore, to present evidence of a physiological relevance of the RelA-NirD in vivo, we decided to follow the growth of WT and ΔnirD mutant during anaerobic glucose fermentation, a condition where the nitrite reductase is expressed but where the substrate is missing. As shown in the new Figure 3E, no difference is observed between WT and ΔnirD mutant under unstressed conditions. However, addition of high concentrations of SMG (which induces isoleucine starvation) significantly increases the growth recovery of a ΔnirD mutant consistent with an elevated level of (p)ppGpp in this strain. To further confirm that the RelA-NirD interaction is important to adjust (p)ppGpp levels under these specific growth conditions, we have created a chromosomal nirDE50K mutant. We first confirmed that this mutant has a functional nitrite reductase activity. Indeed, the nirDE50K mutant grows similarly to the WT in presence of nitrate as sole nitrogen source (new Figure 3—figure supplement 1C). Finally, we observed that nirDE50K mutant displays the same growth defect recovery as the nirD mutant in presence of SMG.

Growth curves of WT cells, Δ*nirD* and *nirDE50K* mutants under anaerobiosis in M9-glucose minimal medium without amino acids (Figure 3E left panel) or supplemented with 1 mM serine, methionine, and glycine (SMG) (Figure 3E right panel) (see Materials and methods). The growth curves are representative of at least 3 independent experiments and the Δ(lag phase) are represented as the means (± SD) of the independent experiments.

Growth curves of WT cells, ΔnirD and nirDE50K mutants under anaerobiosis in defined glycerol minimal medium supplemented with nitrate as the sole electron acceptor and as sole nitrogen source (see Materials and methods and Figure 3—figure supplement 1C). Under nitrate respiration, nitrate is converted to nitrite which is subsequently converted to ammonium (NH4) by the cytosolic NADH-dependent nitrite reductase NirBD. Bacteria can then utilize NH4 as nitrogen source. Therefore, an active NADH-dependent nitrite reductase activity is required for anaerobic growth when nitrate is the sole nitrogen source (Wang et al., 2019).

Therefore, our results present physiological evidence that a specific RelA-NirD interaction can adjust (p)pGpp level in vivo.

A full paragraph explaining these new results is now provided in the revised manuscript.

– Document the ratio of RelA:NirD. Is RelA in a much higher abundance than NirD? How does this relate to the importance of the interaction?

The equilibrium affinity constant (KD) of the RelA-NirD interaction obtained in vitro is around 300 nM and the concentration of RelA in *E. coli* is in the same range (100-400 nM) (Li et al., 2014; Pedersen and Kjeldgaard, 1977). Therefore, at the concentrations above the KD, NirD would interact with RelA and inhibit (p)ppGpp synthetase activity. Under aerobic conditions, NirD is not expressed and concentration is below 20 nM and regulation of RelA by NirD is probably meaningless. These results are consistent with the results presented above. However, under anaerobic growth conditions, NirD is predicted to reache μM range (Khlebodarova et al., 2016) and can therefore saturate RelA to catalytically inhibit its activity. These results are consistent with the observation that deletion of nirD in presence of RelA severely impaired growth recovery under anaerobic glucose fermentation conditions in presence of SMG.

The physiological relevance of the RelA-NirD interaction is now discussed in the revised manuscript.

– Revise the text according to the following remarks:a – Page 3, lines 12-17: with reference to the targets and roles of (p)ppGpp, the authors cite Atkinson et al., 2011; Kanjee et al. 2012; Hauryliuk et al. 2015 and Potrykus and Cashel 2008.Remove here the reference of Atkinson et al., 2011, which described the phylogenic distribution of RSH enzymes.

We agree, the reference has been removed.

– Add to or replace Kanjee et al. 2012 more recent papers in which they identified (p)ppGpp targets: Corrigan et al., 2016 (PMID: 26951678); Zhang et al., 2018 (PMID: 29511080); Wang et al., 2019 (PMID: 30559427).– Add to or replace Hauryliuk et al. 2015 and Potrykus and Cashel 2008 by updated reviews: Hengge 2020 (PMID: 32244175); Irving et al. 2020 (PMID: 33149273).

We thank the referees, changes have been made accordingly.

b – Page 3, Line 23: in relation to the CTD, references Pausch 2020 Cell Reports, Takada 2020 Front Microbiol, and Gratani 2018 Plos Genetics should be included.c – When describing RelA and SpoT as well as the mechanisms regulating their activities, the authors refer to only two research papers (Hogg et al., 2004; Mechold et al., 2002) and again the same reviews (Hauryliuk et al., 2015; Potrykus and Cashel 2008). Add Tamman et al., 2020 (PMID: 32393900).

We apologize, changes have been made accordingly.

d – Page 4, Line 4: two recent reviews extensively describe how stress signals are sensed by RSH enzymes. In place of Potrykus and Cashel, 2008, cite these:Irving SE, Corrigan RM. (PMID: 29493495); Ronneau S, Hallez R. (PMID: 30980074).

We agree, changes have been made accordingly.

e – Page 4, line 24, remove the "in" after Rsd.

We agree, the change has been made accordingly.

f – Page 4: It would be interesting to add that RelA can also be activated by other stressful conditions than aa starvation which ultimately leads to aa starvation, e.g. fatty acid starvation that leads to lysine starvation (Sinha et al., 2019, PMID: 31400173).

We agree, changes have been made accordingly.

g – Comment on why GppA relieves toxicity? pppGpp is less toxic than ppGpp, why?

Cited in the manuscript, Sanyal and Harinarayanan (2020) describe interesting in vivo findings regarding the pivotal role of SpoT and GppA hydrolase in lowering activation of RelA by the alarmone guanosine penta phosphate (pppGpp). Indeed, they present genetic evidence that pppGpp exerts heightened toxicity compare to that of ppGpp. The authors provide a simple model in which toxicity of pppGpp is mediated by its positive feedback activation of RelA (even in absence of activating signal) and thus setting up massive (p)ppGpp accumulation in *E. coli* cells. While the exact molecular mechanisms are not known, we believe that our observations that gppA reverts toxicity associated to (p)ppGpp accumulation are consistent with these findings.

Note also that early on, Kudrin et al. (2018) reported in vitro that RelA’s synthetic activity is adjusted by pppGpp, which stimulates synthesis of ppGpp from GDP to maximize (p)ppGpp accumulation.

h – Page 8, line 20 and page 13, line 9: add reports describing RSH regulation by– the competence proteins ComGA: Hahn et al., 2015 (PMC4722805)– the phosphotransferase system: Karstens et al., 2014 (PMID: 24515609); Ronneau et al., 2016 (PMID: 27109061); Ronneau et al., 2019 (PMID: 30496454).

The referee is correct, changes have been made accordingly.

i – Page 8, line 3, "zing" correct the typo.

We apologize, the change has been made.

j – Page 11, line7 – change to S5B and C to S4B and C. Also include A-E labels in figure S4 and a specific legend for S4E.

We apologize, changes have been made.

k – Page 19 line 15 – change broh to broth.

We apologize, the change has been made.

l – What do you mean by "ribosome-independent inhibition of the stringent response"?

NirD can inhibit the (p)ppGpp synthesis from the constitutively active truncated RelAN-terminal both in vivo and in vitro. Given that the CTD region is responsible for anchoring RelA to the ribosome and that ribosome is missing in our in vitro assay, our results support the notion that inactivation of RelA is possible even when the protein is not associated to the ribosome. Therefore, rather than preventing activation of RelA, our results support a simple model of regulation in which NirD directly interacts with the catalytic domains of RelA to inhibit the synthetic activity.